# Bone marrow adipocytes drive the development of tissue invasive Ly6C<sup>high</sup> monocytes during obesity

Parastoo Boroumand[1,2], David C Prescott[3,4], Tapas Mukherjee[3], Philip J Bilan[1], Michael Wong[1], Jeff Shen[1], Ivan Tattoli[4], Yuhuan Zhou[1], Angela Li[5], Tharini Sivasubramaniyam[5], Nancy Shi[1], Lucie Y Zhu[1], Zhi Liu[1], Clinton Robbins[4,5], Dana J Philpott[3], Stephen E Girardin[3,4], Amira Klip[1,2]*

[1]Cell Biology Program, The Hospital for Sick Children, Toronto, Canada; [2]Department of Biochemistry, University of Toronto, Toronto, Canada; [3]Department of Immunology, University of Toronto, Toronto, Canada; [4]Department of Laboratory Medicine and Pathophysiology, University of Toronto, Toronto, Canada; [5]Toronto General Hospital Research Institute, Toronto, Canada

*For correspondence:
amira@sickkids.ca

**Abstract** During obesity and high fat-diet (HFD) feeding in mice, sustained low-grade inflammation includes not only increased pro-inflammatory macrophages in the expanding adipose tissue, but also bone marrow (BM) production of invasive Ly6C<sup>high</sup> monocytes. As BM adiposity also accrues with HFD, we explored the relationship between the gains in BM white adipocytes and invasive Ly6C<sup>high</sup> monocytes by in vivo and ex vivo paradigms. We find a temporal and causal link between BM adipocyte whitening and the Ly6C<sup>high</sup> monocyte surge, preceding the adipose tissue macrophage rise during HFD in mice. Phenocopying this, ex vivo treatment of BM cells with conditioned media from BM adipocytes or bona fide white adipocytes favoured Ly6C<sup>high</sup> monocyte preponderance. Notably, Ly6C<sup>high</sup> skewing was preceded by monocyte metabolic reprogramming towards glycolysis, reduced oxidative potential and increased mitochondrial fission. In sum, short-term HFD changes BM cellularity, resulting in local adipocyte whitening driving a gradual increase and activation of invasive Ly6C<sup>high</sup> monocytes.

## Editor's evaluation

This is an important study providing compelling observations describing the metabolic response of monocyte subsets in conjunction with crosstalk with adipocytes in the murine bone marrow during leanness and obesity. This work convincingly reports bone marrow adaptations to lipid signals and how obesity affects monocyte biology, recruitment and differentiation locally in the bone marrow. This study at the crossroad of metabolism and immunology will be of interest for a wide community interested in obesity and macrophage biology.

## Introduction

A continuum of chronic obesity leading to metabolic disorders such as insulin resistance and eventually type 2 diabetes is of paramount concern worldwide (*Centers for Disease Control and Prevention, 2017*; *OECD, 2017*; *World Health Organization, 2016*). An important contributor to the development and exacerbation of insulin resistance is a gain in macrophages in adipose tissue (AT) (*Liu et al., 2019*; *Weisberg et al., 2003*), muscle (*Fink et al., 2014*; *Saghizadeh et al., 1996*) and liver (*Lanthier et al., 2010*) as well as their enhanced pro-inflammatory (*Lumeng et al., 2007*; *Lumeng et al., 2008*;

*De Souza et al., 2005*) and glycolytic switch (*Sharma et al., 2020*). This low-grade inflammation is partly explained by circulating monocyte infiltration into tissues where they differentiate to pro-inflammatory macrophages (*Kohyama et al., 2016*).

Ly6C[high] monocytes in mice, and classical CD14[+] CD16[-] monocytes in humans, constitute the predominant tissue-infiltrating monocyte subset during chronic inflammation. These tissue-invasive monocytes are classified by high surface abundance of the Ly6C protein, the chemokine receptor CCR2 and the adhesion molecule L-selectin, all of which allow for monocyte extravasation from the BM and infiltration into the tissues (*Auffray et al., 2007*; *Boniakowski et al., 2018*; *Fink et al., 2014*; *Geissmann et al., 2003*; *Weisberg et al., 2006*). With obesity, the development of Ly6C[high] monocytes along with their stem and progenitor cells in the BM is boosted (*Griffin et al., 2018*; *Manicone et al., 2016*; *Singer et al., 2014*) leading to a rise in available circulating Ly6C[high] monocytes for tissue-infiltration (*Friedrich et al., 2019*; *Singer et al., 2014*).

Various obesity-driven factors amalgamate for greater Ly6C[high] monocyte infiltration into the AT including (a) increased AT-dependent chemoattraction (*Kamei et al., 2006*; *Nagareddy et al., 2014*; *Nov et al., 2013*; *Renovato-Martins et al., 2017*) and (b) enhanced BM myelopoiesis (*Griffin et al., 2018*; *Manicone et al., 2016*; *Singer et al., 2014*). However, it is not clear whether the obesity-induced BM myelopoiesis occurs prior to, concurrently, or after the elevation of AT macrophages. Since the BM is a highly vascularized tissue, home to hematopoietic and mesenchymal lineages, and is dynamically reactive to environmental changes, we reasoned it might be an early responder to HFD-induced obesity, producing increased Ly6C[high] monocytes prior to their accumulation in the AT. Further, since BM-resident adipocytes undergo hyperplasia and hypertrophy during obesity (*Boroumand and Klip, 2020*; *Doucette et al., 2015*; *Scheller et al., 2015*; *Tratwal et al., 2020*) we contemplated that this process might be temporally and potentially causally related to the HFD-driven BM myelopoiesis and Ly6C[high] monocytosis. Finally, we considered whether the response of the monocytic lineage to the BM environment during obesity includes a metabolic shift leading to the increase in Ly6C[high] monocytes. In fact, and surprisingly, little is known about the metabolic orientation of the Ly6C[high] monocytes in states of obesity, although there is mounting evidence of a link between a macrophage-intrinsic metabolic shift and their pro-inflammatory orientation in this condition (*O'Neill et al., 2016*; *Sharma et al., 2020*; *Thapa and Lee, 2019*; *Verdeguer and Aouadi, 2017*).

Hence, the overall hypothesis driving this study is that, with obesity, the increased production of Ly6C[high]-bearing monocytes in the BM is accompanied and possibly caused by a glycolytic shift in these cells, driven by changes in bone marrow adipocytes (BMA). Accordingly, we investigated the temporality of enhanced BM myelopoiesis, BM Ly6C[high] monocyte accumulation, and AT macrophage accrual during high fat diet (HFD) feeding in mice. Next, we explored the potential for a causal relationship between the obesity-induced changes in BMA and increased Ly6C[high] monocyte production. Using an in vitro platform, we deconstructed the effects of obesity-induced BM cellular changes on Ly6C[high] monocyte expansion. We report that BM Ly6C[high] monocytes accumulate in the BM prior to the gain in pro-inflammatory AT macrophages. The rise in BM monocytes temporally correlated with the elevation in BMA. Moreover, BMA whitening correlated with the timely increases in BM Ly6C[high] monocytes, and remarkably, isolated BMA from HFD-fed mice directly prompted Ly6C[high] monocyte development ex vivo. This monocyte developmental cue was also induced ex vivo by *bona fide* white adipocytes while *bona fide* brown adipocytes favoured Ly6C[low] monocyte predominance. Noteworthy, the white adipocyte-induced swap in proportion of Ly6C[low] relative to Ly6C[high] monocytes was preceded by a switch in monocyte metabolism from oxidative to glycolytic. Mechanistically, brown adipocytes increased Ly6C[low] monocyte proliferation while white adipocytes appear to promote Ly6C[low] monocyte conversion and potentially progenitor expansion to increase Ly6C[high] preponderance. We propose that HFD-induced remodelling in BMA disrupts the normal balance of BM monocyte subsets leading to a higher output of tissue invasive Ly6C[high] monocytes.

## Results

### Increased fasting blood glucose, peripheral and BM adiposity by 3 weeks of HFD

To study the temporal relationship of HFD-induced myeloid changes in the BM relative to that in epididymal white AT (EWAT), C57BL/6 mice were fed HFD or control low fat diet (LFD) for either 3, 8,

or 18 weeks. HFD-fed mice had higher body weight and impaired fasting blood glucose by 3 weeks, compared to controls (*Figure 1A and B*). Body weight and hyperglycemia continued to rise with prolonged feeding. HFD-fed mice also displayed higher EWAT weight (*Figure 1C*) and inter-scapular brown AT (BAT) weight at 3 weeks (not shown). Cross-sectional tibia and femur bone histology visualized via H & E and Oil Red O staining showed an increase in BMA size by 3 weeks and an increase in BMA number by 8 weeks of HFD (*Figure 1D–G*). This indicated that 3 weeks of HFD sufficed to induce whole body metabolic disturbance and augmented adiposity in peripheral tissues and in the BM.

## BM monocytes increase prior to accumulation of EWAT macrophages and monocytes with HFD

Although total EWAT weight increased by 3 weeks, myeloid cells in the stromal vascular fraction (SVF) of AT were not elevated until 8 weeks of HFD, as indicated by gene expression of the general myeloid marker *Itgam* (encoding CD11b) and monocyte markers Gr1$^-$ CD115$^+$ CD11b$^+$ Ly6C$^+$ (determined by flow cytometry) (*Figure 2A and D*). Similarly, AT macrophages, assessed by gene expression of the AT macrophage marker *Adgre1* (encoding F4/80) and by flow cytometry counting of CD11b$^+$ Ly6C$^-$ F4/80$^+$ macrophages in the SVF were not elevated at 3 weeks of HFD but were at 8 weeks (*Figure 2B and E*). Further, gene expression of the pro-inflammatory myeloid marker *Itgax* (encoding CD11c) and surface detection of the pro-inflammatory markers CD11b$^+$ Ly6C$^-$ F4/80$^+$ CD11c$^+$ were also higher in the SVF by 8 weeks of HFD relative to control diet, but not at 3 weeks (*Figure 2C and F*). Of note, gene expression of the embryonically-conserved tissue-resident AT macrophage markers *Timd4* and *Marco* (*Beattie et al., 2016*; *Gibbings et al., 2015*) was not elevated at 3 or at 8 weeks and only rose at 18 weeks of HFD (*Figure 2G–H*). This indicates that the gain in AT macrophages at 8 weeks of HFD is unlikely due to proliferation of resident macrophages and instead likely results from monocyte recruitment from the circulation.

In the BM, total monocytes, gated via the Gr1$^-$ CD115$^+$ CD11b$^+$ Ly6C$^+$ markers, increased at 3 weeks and continued to rise by 8 and 18 weeks of HFD (*Figure 2I*). Using the surface abundance of the Ly6C protein as well as the CX3CR1 and CCR2 markers (*Auffray et al., 2007*; *Auffray et al., 2009*; *Kratofil et al., 2017*), we subcategorized Ly6C$^{low}$ and Ly6C$^{high}$ monocyte subsets. The two populations were identified by FACS analysis using the gating strategy illustrated in *Figure 2—figure supplement 1*. At 3 weeks into the feeding study, Ly6C$^{low}$ and Ly6C$^{high}$ monocytes represented about 40% of the total BM monocyte population in the control diet group, whereas HFD promoted a higher proportion of Ly6C$^{low}$ monocytes (*Figure 2J*). However, by 8 weeks of HFD there was a marked shift, whereby the relative proportion of Ly6C$^{low}$ monocytes in the total monocyte population was *lower* and that of Ly6C$^{high}$ monocytes was *higher* than under the control diet (*Figure 2J*). These differences were more prominent by 18 weeks of HFD feeding (*Figure 2J*). Collectively, these findings revealed that, in the BM, HFD favoured monocytosis at 3 weeks, with a subsequent and prominent switch towards tissue invasive Ly6C$^{high}$ monocytes at 8 weeks that continued into 18 weeks of HFD. *Singer et al., 2014* also demonstrated an increase in BM monocytosis and elevated monocyte progenitor cells with 18 weeks of HFD, that persisted after BM transplantation into chow-fed recipients. In the present study, while we clearly recorded an earlier and continuing rise in total monocytes in HFD-fed mice (*Figure 2I*), we did not observe statistically significant increases in monocyte progenitor and stem cells at any time point analyzed (*Figure 2—figure supplement 2A-C*).

## Glycolytic metabolic reprograming in monocyte subsets with HFD

Recent discoveries have highlighted the importance of glycolytic metabolism in the activation of AT macrophages with HFD (*Lee et al., 2019*), but the metabolic status of monocytes in obesity has not been reported, although some studies have investigated the metabolism of circulating human CD14$^+$CD16$^-$ monocytes (homologous to murine tissue invasive Ly6C$^{high}$ monocytes) during aging (*Lee et al., 2019*; *Pence and Yarbro, 2019*). To assess whether HFD affects the metabolic status of BM monocyte subsets in obesity, we analyzed Ly6C$^{low}$ and Ly6C$^{high}$ from control and HFD mice at 3, 8, and 18 weeks of feeding. The mitochondrial activity-indicator dye MitoTracker Red CMXRos was incorporated into the monocyte flow cytometry panel. MitoTracker Red CMXRos is recruited to the mitochondrial membrane proportionally to the degree of mitochondrial membrane potential and is thus an indicator of mitochondrial activity in oxidative phosphorylation. The results exhibit that Ly6C$^{low}$ monocytes have a higher mitochondrial membrane potential than Ly6C$^{high}$ monocytes in the

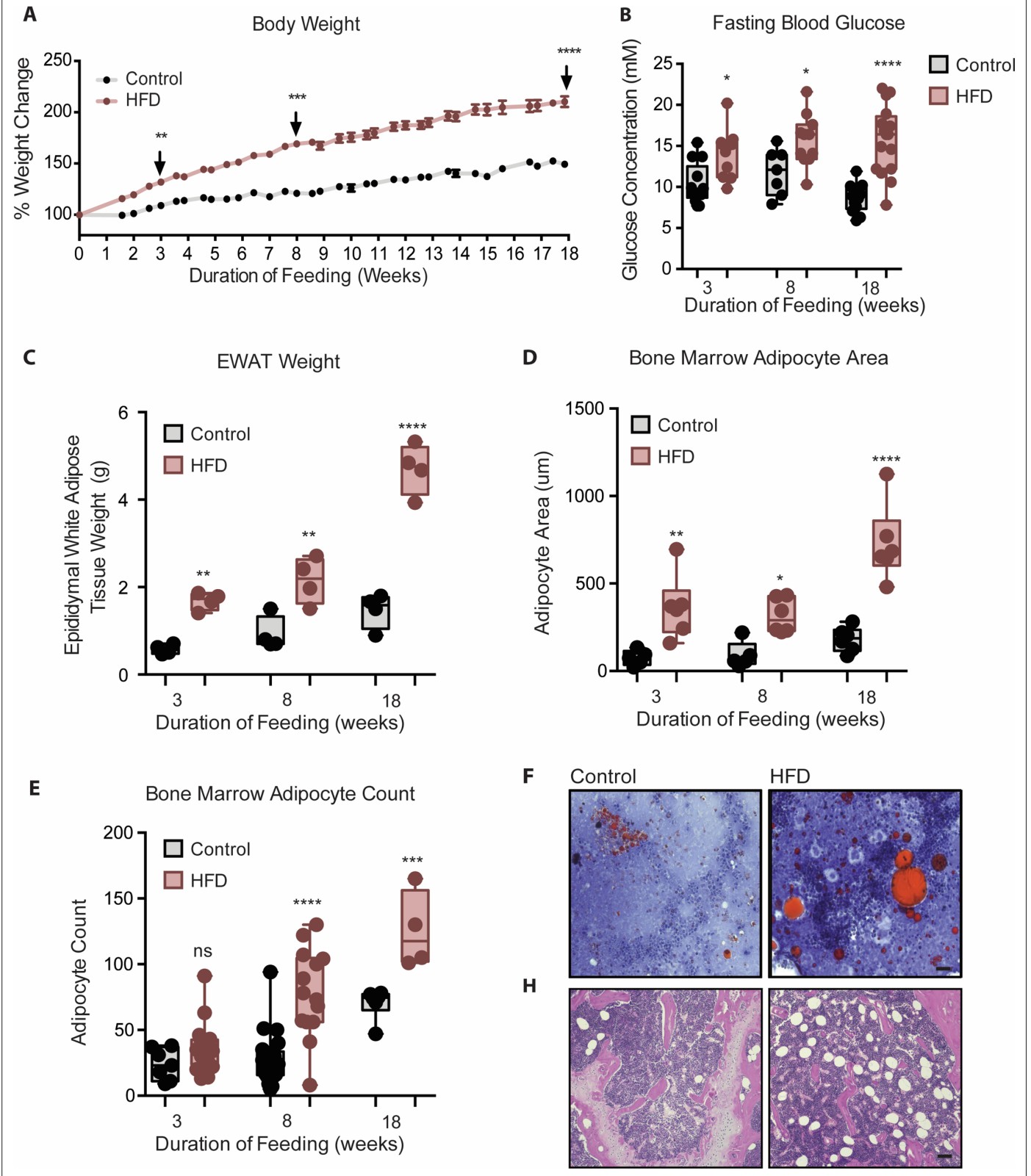

**Figure 1.** Mice fed HFD increased body weight, fasting blood glucose, visceral and bone marrow adipose tissue. (**A**) Body weight was measured twice a week throughout the course of feeding (n=14 per dietary group). (**B**) Mice were fasted for 4 hr after which fasting blood glucose was measured (n=9–14 per dietary group). (**C**) EWAT was weighed the day of sacrifice (n=4 per dietary group at each feeding duration). (**D**) Area covered by BM adipocytes were measured following H&E to visualize adipocyte ghosts using quantified using ImageJ software (n=6 mice in each dietary group at each feeding

*Figure 1 continued on next page*

*Figure 1 continued*

duration). (**E**) BM adipocyte numbers per field were blindly counted following H&E staining using ImageJ software version 1.53. (n=4–14 mice in each dietary group per feeding duration). (**F**) Representative images of femurs from mice fed control and HFD diet for 3 weeks, that were fixed in formalin for 2 days followed by 14 days EDTA decalcification and subsequently stained with Oil Red O (ORO). (**G**) Representative images of femurs from mice fed control and HFD diet for 8 weeks, that were fixed in formalin for 2 days followed by 14 days EDTA decalcification and subsequently stained with H&E. Scale bar is 25 µm. Results are expressed as mean ± SEM, n=4–14 mice per group. One-way ANOVA was used for statistical analysis at each dietary group and across feeding periods. *p<0.05, **p<0.01, ***p<0.001, ****p<0.0001 and ns = not significant p>0.05.

The online version of this article includes the following source data for figure 1:

**Source data 1.** Mice fed HFD increased body weight, fasting blood glucose, visceral and bone marrow adipose tissue.

control mice (*Figure 3A–C*). Strikingly, 3 weeks of HFD reduced the mitochondrial membrane potential in both monocyte subsets indicated by the left-ward shift in the MitoTracker Red CMXRos peak (*Figure 3A*). The HFD-induced effects were more pronounced by 8 and 18 weeks of HFD (*Figure 3B and C*). To further understand the effects of HFD on the developing BM monocytes, Ly6C$^{low}$ and Ly6C$^{high}$ monocytes from mice fed control or HFD for 3, 8, or 18 weeks were isolated via fluorescent activated cell sorting (FACS) and plated for metabolic analysis with a Seahorse XFe96 Analyzer. The results revealed a gradual lowering in the oxygen consumption rate (OCR) of Ly6C$^{low}$ monocytes with HFD (*Figure 3D–I*). Comparison of the area under the curve (AUC) showed that at 18 weeks, the OCR of the Ly6C$^{low}$ monocytes was significantly lowered by HFD (*Figure 3I*). Interestingly, at 8 and 18 weeks, the AUC of the OCR of the Ly6C$^{low}$ monocytes was significantly higher than the AUC of the OCR of Ly6C$^{high}$ monocytes in the control group (*Figure 3H–I*). Moreover, this difference was lost in the HFD group as the AUC of the OCR of Ly6C$^{low}$ monocytes diminished (*Figure 3H–I*).

A comparison of the various sections of the OCR graph illustrated that the changes in OCR observed with 3 weeks of HFD were not manifest at basal respiration nor at the ATP-dependent respiration, rather they were evident in the maximal respiratory capacity portion of the metabolic stress test (*Figure 3D*). A drop in basal respiration of both monocyte subsets emerged with 8 weeks of HFD and became more pronounced with 18 weeks of HFD (*Figure 3E–F*). This showcased the sustained effects of HFD on monocyte metabolic status. Conversely, the extracellular acidification rate (ECAR) of Ly6C$^{low}$ monocytes was higher than that of Ly6C$^{high}$ monocytes, consistent with a more active metabolism in the former (*Figure 3J*). However, this difference was lost with HFD, as ECAR dropped in the Ly6C$^{low}$ monocytes and rose in the Ly6C$^{high}$ monocytes relative to the control diet group (*Figure 3J*). Overall, these results suggest that, along with promoting the preponderance of Ly6C$^{high}$ monocytes, HFD also lowers OCR in Ly6C$^{low}$ monocytes.

Endorsing these results, an analysis of the BM showed elevated lactate levels across all HFD time points (*Figure 3K*). This increase in lactate may be due to the glycolytic shift in BM monocytes described above, among other potential sources within the BM cellular composition. Further analysis of the BM soluble fraction failed to evince any changes between control and 18-week HFD-fed mice in pro-inflammatory cytokines *Tnfα, Il1β, Il6, Cxcl1 or Ccl2* (*Figure 3—figure supplement 1A-C*). Similarly, there were no differences between diet groups in pro-inflammatory gene expression of *Il6, Il1β* or *Itgax* in FACS-sorted Ly6C$^{low}$ and Ly6C$^{high}$ monocytes (Not Shown). However, *Tnfα* gene expression was higher in FACS-sorted Ly6C$^{high}$ monocytes from 18 week HFD-fed mice compared to controls (*Figure 3—figure supplement 1D*). Collectively, these findings illustrate that HFD alters the BM metabolic micro-environment with little effect on pro-inflammatory cytokine secretion (based on the panel tested).

## Expression of mitochondrial fusion and fission genes in monocytes is altered by HFD

The efficiency of mitochondrial function is intrinsically reliant on its structure. Mitochondria are highly dynamic in structure, largely regulated by concentration of proteins that control fission and fusion (*Gottlieb and Bernstein, 2017*). Dynamic transitions are closely aligned to metabolic performance. To further assess the mitochondrial activity of monocytes, we enquired whether their expression of mitochondrial dynamic genes is altered with HFD compared to the control group.

The analyzed genes involved in mitochondrial fusion were *Mfn1, Mfn2, Opa1, Tomm20,* and *Tomm40*, while those involved in mitochondrial fission were *Drp1, Ppid (CypD),* and *Fis1*. BM monocytes from mice fed HFD for 3 weeks did not present with differences in gene expression in the tested

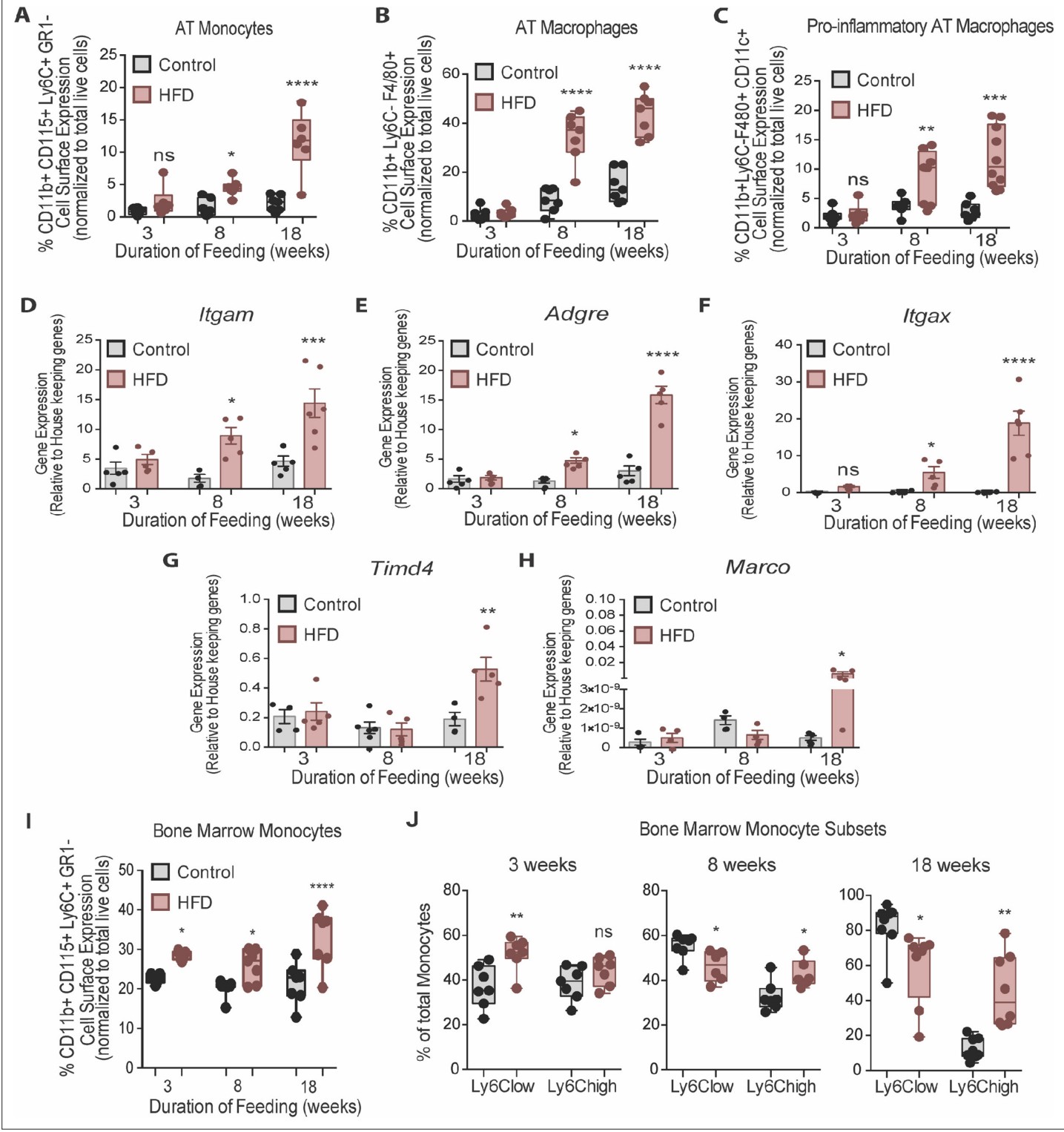

**Figure 2.** Bone marrow monocytes increased prior to adipose tissue macrophages. Analysis of monocytes and macrophages in the stromal vascular fraction (SVF) of adipose tissue via flow cytometry analysis, identified first by size and granularity through forward and side scatter and then using the following markers: CD11b CD115 F4/80 Ly6C GR1 CD11c (n=7–10 mice in each dietary group at each feeding duration). (**A**) Monocytes, (**B**) Total macrophages, (**C**) Pro-inflammatory macrophages. (**D–F**) Whole adipose tissue gene expression of the myeloid marker *Itgam*, macrophage marker *Adgre* and pro-inflammatory myeloid marker *Itgax* by RTqPCR. Each symbol represents a biological replicate analyzed in duplicate and normalized to average of a group of housekeeping genes. (n=5 mice in each dietary group at each feeding duration). (**G–H**) Gene expression analysis of embryonic

*Figure 2 continued on next page*

*Figure 2 continued*

derived tissue resident macrophage markers *Timd4* and *Marco* by RTqPCR. Each symbol represents a biological replicate analyzed in duplicate and normalized to average of a group of housekeeping genes (n=3–5 mice in each dietary group at each feeding duration). (**I**) Flow cytometry analysis of total BM monocytes (n=7 mice in each dietary group at each feeding duration). (**J**) Monocyte subsets in the bone marrow with HFD-feeding (n=7 mice in each dietary group at each feeding duration). Results are expressed as mean ± SEM. One-way ANOVA was used for statistical analysis at each dietary group and across feeding periods. *p<0.05, **p<0.01, ***p<0.001, ****p<0.0001 and ns = not significant p>0.05.

The online version of this article includes the following source data and figure supplement(s) for figure 2:

**Source data 1.** Bone marrow monocytes increased prior to adipose tissue macrophages.

**Figure supplement 1.** Flow cytometry gating strategy for identification of Ly6C^low^ and Ly6C^high^ monocytes and their progenitor cells.

**Figure supplement 2.** Monocyte progenitors and stem cells were not significantly altered with HFD in vivo.

**Figure supplement 2—source data 1.** Monocyte progenitors and stem cells were not significantly altered with HFD in vivo.

mitochondrial fusion genes compared to control diet counterparts (*Figure 4A*). BM monocytes isolated from mice fed HFD for 8 weeks showed downregulated *Mfn1* and *Mfn2* compared to the controls (*Figure 4B*). BM monocytes isolated from 18-week HFD-fed mice downregulated the mitochondrial fusion genes *Mfn2*, *Opa1*, and *Tomm20* (*Figure 4C*). Furthermore, from the tested mitochondrial fission genes, *Drp1* was upregulated in the BM monocytes of 3 weeks HFD-fed mice (*Figure 4D*). *Drp1* and *Ppid* were also upregulated in BM monocytes of 8 weeks HFD-fed mice (*Figure 4E*). At 18 weeks, all three tested mitochondrial fission genes, *Drp1*, *Ppid*, and *Fis1* were upregulated (*Figure 4F*). This gene expression pattern suggests that HFD skews the monocyte gene expression to favour downregulation of genes involved in mitochondrial fusion and upregulation of genes involved in fission.

Given that Ly6C^low^ and Ly6C^high^ monocytes presented with differences in metabolic preference (oxidative and glycolytic, respectively), and the known link of glycolytic metabolism with mitochondria fragmentation and conversely oxidative metabolism to mitochondrial fusion, we assessed whether the expression patterns of the genes governing mitochondrial fission and fusion would correlate distinctly with the BM monocyte subset abundance in control and HFD-fed mice. Indeed, and notably, the abundance of the Ly6C^high^ monocyte subset negatively correlated with *Mfn2* and positively correlated with *Drp1* levels (*Figure 4G–H*). In contrast, the abundance of BM Ly6C^low^ monocytes did not correlate with the expression patterns of either *Mfn2* or *Drp1* (*Figure 4G–H*). These data are consistent with our metabolic findings suggesting Ly6C^high^ monocytes are more glycolytic and adopt a gene expression pattern that would promote mitochondrial fragmentation.

## A shift in BMA type at 8 weeks of HFD influences monocyte subset preference and mitochondrial dynamic gene expression

Hematopoietic myeloid cells reside in specific niches of the BM that determine their cellular interaction with other co-resident BM cells (*Boroumand and Klip, 2020*). BMA, while representing a unique type of adipocytes, respond to obesity with an increase their white adipocyte-like fat storing properties and a decrease in brown-like characteristics (*Ambrosi et al., 2017*; *Doucette et al., 2015*; *Krings et al., 2012*; *Scheller et al., 2016*). To explore the temporal relation of BMA whitening in relation to the development of monocyte subsets, we followed the gene expression of a suite of recently identified markers that are good differentiators of adipocyte types (*Nascimento et al., 2014*; *Ussar et al., 2014*) in isolated BMA from femur and tibia of control and HFD-fed mice. By 3 weeks into HFD, the white adipocyte marker Asc1 (encoding the guanine nucleotide-binding protein subunit beta-like protein) was significantly higher and the brown adipocyte marker P2*rx*5 (encoding purinergic receptor P2X) trended to be lower albeit without reaching statistical significance, relative to control diet. White adipocyte markers *Lep* (leptin) and *Adipoq* (adiponectin) as well as the brown adipocyte markers *Ucp1* (UCP1) and *Cdc42* (*Pat2*, Integrin alpha pat-2) did not change (*Figure 5A*). By 8 weeks of HFD, *Asc1* mRNA remained elevated and *Lep* and *Adipoq* trended in the same direction (*Figure 5B*). Brown adipocyte markers *P2rx5* and *Ucp1* in the isolated BMA of HFD mice were markedly lower relative to those from control diet mice (*Figure 5B*). Thus, 8 weeks of HFD sufficed to shift BMA towards white adiposity. By 18 weeks of HFD, all white adipocyte markers genes showed were upregulated while brown adipocyte markers were downregulated compared to the control group (*Figure 5C*). Analyses of adipocyte differentiation and progenitor cell markers were conducted in the BM pelleted fraction. *Hoxc8*, a regulator of adipocyte progenitors (*Nakagami, 2013*), was

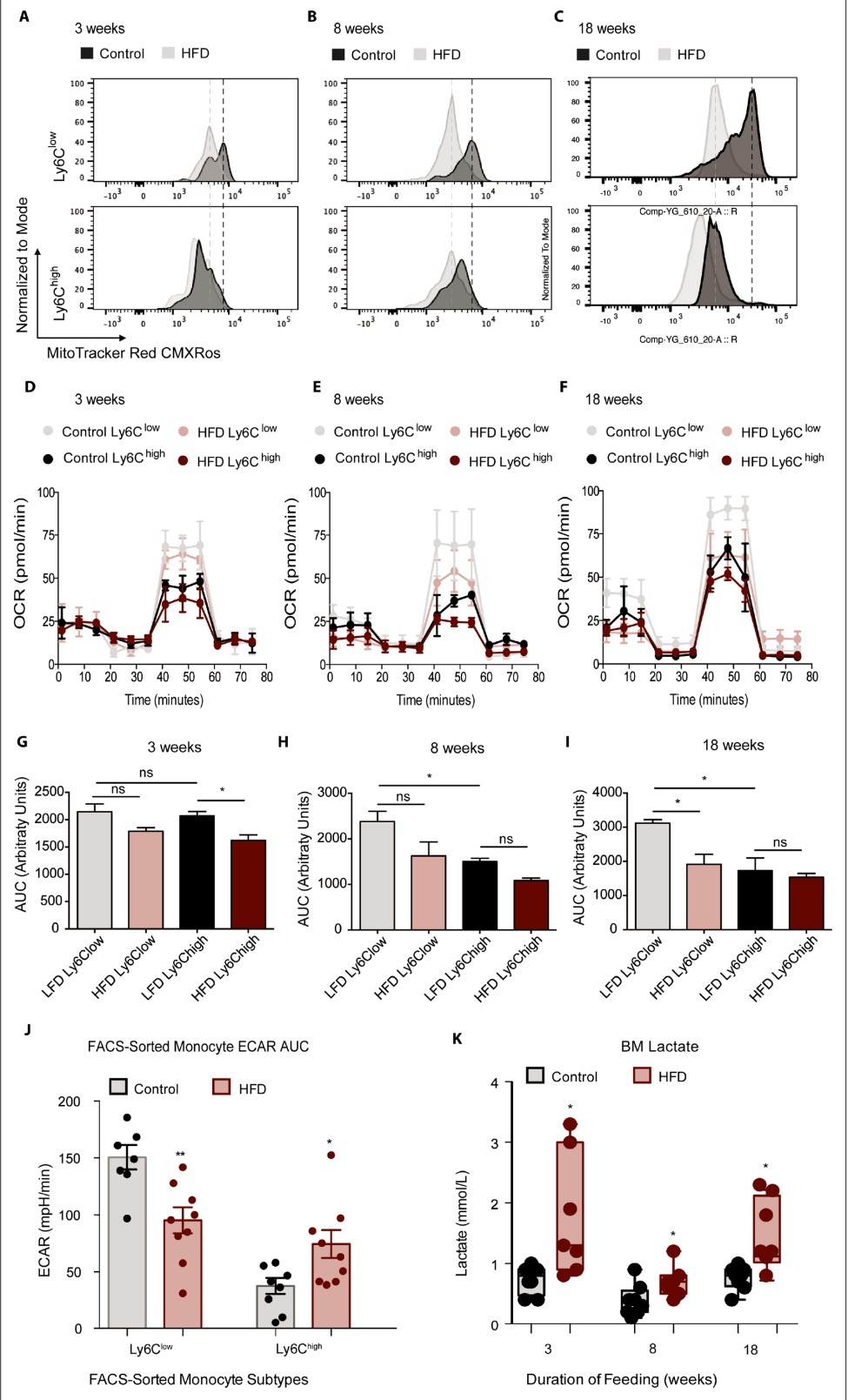

**Figure 3.** HFD blunted the mitochondrial oxidation of monocyte subsets. (**A–C**) Bone marrow cells from mice fed control or HFD for 3, 8, or 18 weeks (n=3 mice per dietary group) were incubated with Mitotracker Red CMXRos for 17 min at 37 °C. This is a fixable red-fluorescent dye that stains mitochondria dependent on their membrane potential. Cells were then fixed, stained with fluorescent antibodies for flow cytometry, and subjected

*Figure 3 continued on next page*

*Figure 3 continued*

to fluorescent activated cell sorting (FACS) to identify Ly6C$^{low}$ and Ly6C$^{high}$ monocyte populations. (**D–F**) Ly6C$^{low}$ and Ly6C$^{high}$ monocytes from mice fed control or HFD for 3, 8, or 18 weeks, (n=3 mice per dietary group) were isolated via FACS and plated for metabolic analysis of their Oxygen Consumption Rate (OCR) with a Seahorse XFe96 Analyzer. (**G–I**) Area under the curve of OCR of FACS-sorted Ly6Clow and Ly6Chigh monocytes from control and HFD-fed mice measured by the Seahorse Assay (n=3 mice per dietary group). (**J**) Area under the curve of ECAR measurement from FACS-sorted Ly6C$^{low}$ and Ly6C$^{high}$ monocytes of control and HFD-fed mice measured by the Seahorse Assay and pooled from all feeding durations. (**K**) BM lactate measurement of the supernatant fraction in HFD and control samples quantified using a lactate Plus Meter and normalized to the initial volume of BM isolated from the femur (n=7 mice per dietary group). Results are expressed as mean ± SEM. One-way ANOVA was used for statistical analysis. *p<0.05, **p<0.01 and ns = not significant p>0.05. Graph in panels C was created using FlowJo software (Becton, Dickinson Company).

The online version of this article includes the following source data and figure supplement(s) for figure 3:

**Source data 1.** HFD blunted the mitochondrial oxidation of monocyte subsets.

**Figure supplement 1.** BM pro-inflammatory factors did not change with HFD.

**Figure supplement 1—source data 1.** BM pro-inflammatory factors did not change with HFD.

---

significantly upregulated at 8 weeks along with *Hoxc9* at 18 weeks of HFD (***Figure 5—figure supplement 1B,C***). The beige adipocyte progenitor marker *Tmem26* was not significantly higher at any of the tested HFD durations (***Figure 4A-C***). Expression of the brown adipocyte determinant genes *Tbx1* and *Lhx8* was undetectable in the isolated BMA. Thus, during the course of HFD, while markers for white adipocyte progenitor cells were upregulated, precursors of beige/brown adipocytes were not affected.

The rise in BM Ly6C$^{high}$ monocytes and the shift in BMA whitening both manifested by 8 weeks of HFD. Correlative analysis was performed for both Ly6C$^{low}$ and Ly6C$^{high}$ with the white adipocyte marker *Asc1* and the brown adipocyte marker *Ucp1*. The results evinced a significant positive correlation (p value of 0.0123) for Ly6C$^{high}$ monocytes with *Asc1* gene expression (***Figure 5D***). Conversely, the surge in Ly6C$^{high}$ monocytes presented a significant negative correlation (p value of 0.0415) relative to *Ucp1* gene expression (***Figure 5E***). Changes in Ly6C$^{low}$ monocytes did not significantly correlate with *Asc1* (p value of 0.0768) or *Ucp1* (p value of 0.1299) (***Figure 5D–E***). Collectively, the above data suggest the possibility that the mere shift in BMA towards white adiposity may directly favour induction of BM Ly6C$^{high}$ monocytes. To test this hypothesis, we generated conditioned media (CM) from BMA (BMA-CM) pooled from three control and three HFD-fed mice, collected 3, 8, and 18 weeks into the diets. In parallel, BM of control mice was used to generate monocytes through a 5-day in vitro differentiation protocol (as reported by ***Francke et al., 2011***, see Methods). These monocytes were treated with BMA-CM from control and HFD-fed mice for 18 hr thereafter. Subsequently, the monocyte population was analyzed via flow cytometry. BMA-CM from 3 weeks HFD-fed mice had no effect on the monocyte subset proportions compared to BMA-CM from control diet-fed mice (***Figure 5F***). Strikingly, however, BMA-CM from 8-week HFD-fed mice applied to BM-derived monocytes from control mice resulted in a significant production of Ly6C$^{high}$ monocytes compared to the effect of BMA-CM from mice on the 8-week control diet (***Figure 5G***). This skewing was further enhanced when exposing the control BM cells to BMA-CM from 18-week HFD-fed mice (***Figure 5H***).

Lastly, we tested whether BMA-CM was capable to induce changes in gene expression of the mitochondrial fusion and fission genes in monocytes, as observed with monocytes isolated from HFD-fed mice. Accordingly, in vitro-generated monocytes were treated with BMA-CM from control and HFD-fed mice. As before, the analyzed genes involved in mitochondrial fusion were *Mfn1*, *Mfn2*, *Opa1*, *Tomm20,* and *Tomm40*, while the tested genes involved in mitochondrial fission were *Drp1*, *Ppid*, and *Fis1*. Monocytes treated with BMA-CM from HFD-fed mice for 18 weeks showed significant downregulated mitochondrial fusion genes *Mfn2*, *Tomm20,* and *Tomm40* (***Figure 5I***). Conversely, the mitochondrial fission genes *Drp1* and *Ppid* were significantly up-regulated in monocytes treated with HFD-derived BMA-CM (***Figure 5J***).

These results not only show that HFD-derived BMA-CM can bias monocyte subset cellularity but also can alter mitochondrial morphology via changes in mitochondrial dynamic gene expression. These changes can have consequential effects on the metabolic properties on the monocytes.

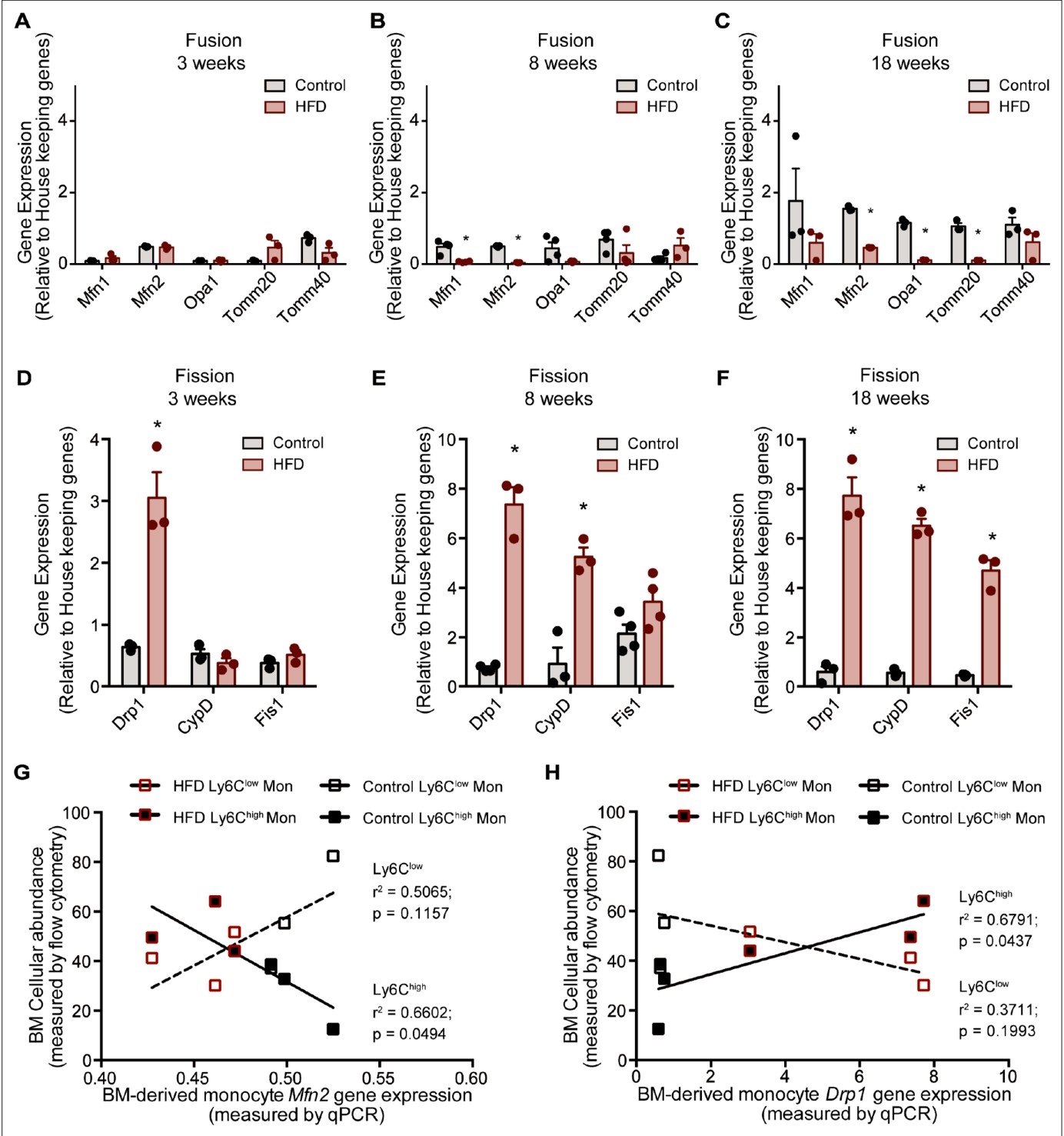

**Figure 4.** HFD downregulated expression of mitochondrial fusion genes and upregulated mitochondrial fission genes. Gene expression of mitochondrial fusion genes *Mfn1, Mfn2, Opa1, Tomm20,* and *Tomm40* in monocytes isolated from control and HFD-fed mice for (**A**) 3, (**B**) 8, (**C**) 18 weeks. Gene expression of mitochondrial fission genes *Drp1*, *Ppid* and *Fis1* in monocytes isolated from control and HFD-fed mice for (**D**) 3, (**E**) 8, (**F**) 18 weeks (n=3 mice per dietary group and samples were analyzed in duplicates). (**G**) Correlative analysis of monocyte subset BM cellular abundance and BM-derived monocyte *Mfn2* gene expression. (**H**) Correlative analysis of the average of monocyte subset BM cellular abundance (n=8 per dietary group) and BM-derived monocyte *Drp1* gene expression (n=3 per dietary group) per dietary group at each time point. Correlative analysis statistics were conducted using the linear regression analysis testing whether slopes and intercepts are significantly different via GraphPad Prism version 6

*Figure 4 continued on next page*

*Figure 4 continued*

(GraphPad Software). The r² represents the goodness of the fit while the P values greater than 0.05 represent whether the slope of the line of best fit is significantly non-zero. Results are expressed as mean ± SEM. One-way ANOVA was used for statistical analysis. *p<0.05 and ns = not significant p>0.05.

The online version of this article includes the following source data for figure 4:

**Source data 1.** HFD downregulated expression of mitochondrial fusion genes and upregulated mitochondrial fission genes.

## White adipocytes favour Ly6C^high monocytes and promote mitochondrial fragmentation

To assess whether the effects of BMA on monocyte subset skewing and metabolic differences can be attributed to BMA whitening, we designed a proof-of-concept experiment studying the influence of *bona fide* white or brown adipocytes on monocyte subset determination and metabolic changes ex vivo. Isolated adipocytes from EWAT and brown adipocyte tissue (BAT) of three mice were pooled to generate white AT adipocytes (WATA) and brown AT adipocytes (BATA), respectively. Confirming the phenotype of the isolated cells, classical white adipocytes markers were solely expressed in the isolated WATA, and classical brown adipocyte markers were exclusively expressed in the BATA (*Figure 5—figure supplement 1D,E*). The isolated WATA and BATA were cultured for 18 h and the corresponding CM were collected for subsequent treatment of monocyte preparations. In vitro-generated BM-derived monocytes from chow-fed mice were treated with WATA-CM, BATA-CM or left untreated. The proportion of Ly6C^low and Ly6C^high monocyte subsets was determined by flow cytometry analysis (*Figure 2—figure supplement 1*). This in vitro paradigm allowed for direct mechanistic analysis of the influence of white or brown adipocytes on monocytes. These results indicate that WATA-CM favoured Ly6C^high monocyte predominance, while BATA-CM prompted Ly6C^low monocyte abundance (*Figure 6A–B*), presenting a novel, differential influence of white and brown adipocytes on monocyte orientation preference.

Since segregated Ly6C^low and Ly6C^high monocyte subsets had different metabolic orientations, we investigated if the differential adipocyte influence on monocyte subset development corresponded to a change in their metabolic status. To this end, in vitro-generated monocytes were treated with WATA-CM, BATA-CM or left untreated for the last 18 hr of the monocyte differentiation protocol. The mitochondrial membrane potential was assessed using MitoTracker Red CMXRos and flow cytometry analysis. Remarkably, Ly6C^low monocytes incubated in vitro with WATA-CM experienced a left-ward shift in MitoTracker Red CMXRos peak indicating a reduction in mitochondrial membrane potential induced by the WATA-CM (*Figure 6C*), reproducing what was observed with monocytes from HFD-fed mice (see *Figure 3A–C*). Furthermore, the gene expression of mitochondrial fusion genes *Mfn1*, *Mfn2*, *Tomm20,* and *Tomm40* were reduced in monocytes treated with WATA-CM (*Figure 6D*). The mitochondrial fission genes *Drp1*, *Ppid* and *Fis1* were up-regulated in monocytes treated with WATA-CM (*Figure 6E*). Conversely, treating monocytes with BATA-CM downregulated the mitochondrial fission gene *Ppid* (*Figure 6E*). To explore if the changes in mitochondrial fusion and fission gene expression correlated with the corresponding mitochondrial morphology, we utilized super-resolution microscopy to image the mitochondrial protein TOM20 in in vitro monocytes adhered to coverslips.

WATA-CM induced a striking increase in the percentage of cells with fragmented mitochondria, from 22% in control CM-treated cells to 64%. On the other hand, BATA-CM raised mitochondrial elongation from 20% in CM-treated cells to 55% (*Figure 6F–H*). These observations suggest that white adipocyte products alter the metabolic status of Ly6C^low monocytes by reducing their mitochondrial membrane potential and altering mitochondrial dynamic gene expression and morphology, thereby linking changes in BMA whitening to monocyte metabolic status. Moreover, the shift to mitochondrial fragmentation coincided with the higher abundance Ly6C^high monocytes resulting from in vitro-generated monocytes with WATA-CM (*Figure 6A and B*).

## White and brown adipocytes influence monocytes through distinct mechanisms

Monocyte precursor cells in the BM in the tissues maintain cell numbers by cell proliferation (*Cassado et al., 2015*). Although we had not detected HFD-induced changes in hematopoietic stem and monocyte progenitor cells in vivo, we assessed the abundance of these progenitor cell populations upon exposing in vitro-generated monocytes to WATA-CM and BATA-CM. We reasoned that the ex vivo

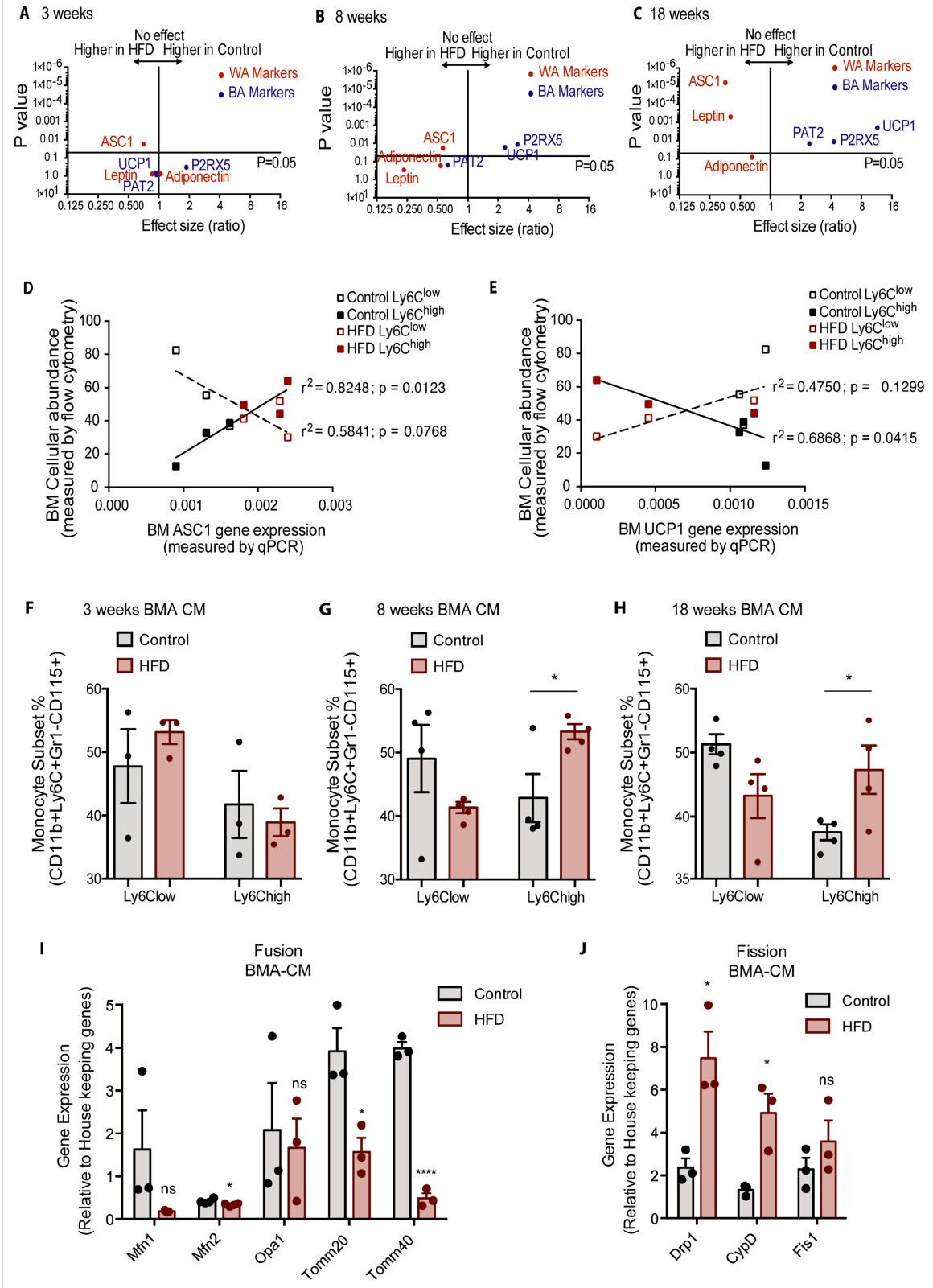

**Figure 5.** Bone marrow adipocyte whitening detected at 8 and 18 weeks of HFD favours increased bone marrow Ly6C$^{high}$ monocytes and induced changes in mitochondrial dynamic gene expression. Volcano plots showing the gene expression of white and brown adipocyte markers in red and blue respectively (**A**) at 3, (**B**) 8, and (**C**) 18 weeks of HFD (n=3–9 per dietary group). Correlative graphs of the (**D**) white adipocyte marker ASC1 (n=8–9 per dietary group) and (**E**) brown adipocyte marker UCP1 (n=6 per dietary group) with Ly6Clow or Ly6Chigh monocyte subset abundance in the BM (n=8).

*Figure 5 continued on next page*

*Figure 5 continued*

Correlative analysis statistics were conducted using the linear regression analysis testing whether slopes and intercepts are significantly different via GraphPad Prism version 6 (GraphPad Software). The $r^2$ represents the goodness of the fit while the p values greater than 0.05 represent whether the slope of the line of best fit is significantly non-zero. (**F–H**) Flow cytometry quantification of naïve Ly6C$^{low}$ and Ly6C$^{high}$ monocytes (n=3–4 per dietary group) from in vitro-generated monocytes pre-treated for 18 hr with pooled BMA-CM collected from three mice fed the control and HFD for 3, 8, and 18 weeks. (**I**) Gene expression of mitochondrial fusion genes *Mfn1, Mfn2, Opa1, Tomm20,* and *Tomm40* in naïve monocytes treated with BMA-CM from control and HFD-fed mice (n=3 per dietary group and samples were run in duplicates). (**J**) Gene expression of mitochondrial fission genes *Drp1, Ppid,* and *Fis1* in naïve monocytes treated with BMA-CM from control and HFD-fed mice (n=3 per dietary group and samples were run in duplicates). Results are expressed as mean ± SEM. One-way ANOVA was used for statistical analysis at each dietary group and across feeding periods. *p<0.05, ****p<0.0001 and ns = not significant >0.05.

The online version of this article includes the following source data and figure supplement(s) for figure 5:

**Source data 1.** Bone marrow adipocyte whitening detected at 8 and 18 weeks of HFD favours increased bone marrow Ly6C$^{high}$ monocytes and induced changes in mitochondrial dynamic gene expression.

**Figure supplement 1.** BMA precursor cell markers change with HFD and phenotyping WATA and BATA.

**Figure supplement 1—source data 1.** BMA precursor cell markers change with HFD and phenotyping WATA and BATA.

experimental setup lent itself well to gain understanding of the mechanisms whereby monocytes undergo changes in subset proportion in response to *bona fide* white and brown adipocyte media. In *Figure 7—figure supplement 1*, the abundance of hematopoietic stem and monocyte progenitor cells (CD117+ cells) was determined using appropriate surface markers (*Table 1*) and flow cytometry analysis as shown (based on *Figure 2—figure supplement 1D*). Surveying several cell populations, only a small gain in myeloid progenitor cell (CMP) was observed in response to WATA-CM treatment (*Figure 7—figure supplement 1A-C*). GMP cells are downstream of CMP during lineage progression of progenitor cells towards monocytes or erythroid cells (*Yáñez et al., 2017*), and so it is noteworthy these cells did not become more abundant with WATA-CM treatment, even though content of CMP, their upstream multipotential progenitors, was slightly elevated (*Figure 7—figure supplement 1C*).

Next, we asked whether proliferation might contribute to the higher proportion of terminally differentiated Ly6C$^{high}$ or Ly6C$^{low}$ monocyte subsets observed following treatment with WATA-CM (in addition to the conversion of Ly6C$^{low}$ into Ly6C$^{high}$) or BATA-CM, respectively. To this end, we treated BM-derived in vitro-generated monocytes with WATA-CM, BATA-CM or left them untreated for 18 hr, with or without the presence of bromodeoxyuridine (BrdU), an indicator of S-phase. Using flow cytometry analysis, DNA-incorporated BrdU was detected in the Ly6C$^{low}$ or Ly6C$^{high}$ monocyte gates of the CD11b versus Ly6C plots, with BrdU as the third colorimetric axis (*Figure 7A*). Although at the highest levels the CD11b signal created a green background signal in the BrdU channel (as seen in samples not treated with BrdU), the actual BrdU signal could be readily distinguished by the yellow to red signal corresponding to the gradient scale shown. Proliferating cells were quantified as percentage of CD11b$^+$ Ly6C$^+$ cells that were positive for BrdU incorporation (yellow to red signal). WATA-CM-treated monocytes rendered the expected increase in Ly6C$^{high}$ monocytes, yet these cells did not show significantly higher incorporation of BrdU than untreated controls, implying that they are not actively replicating their DNA (*Figure 7A,B*). In contrast, BATA-CM increased DNA-incorporated BrdU in the sorted Ly6C$^{low}$ monocyte subset, visualized as yellow to red hues in the CD11b$^+$ Ly6C$^{low}$ cell gate (*Figure 7A*), and the number of proliferating cells was elevated by 25-fold (*Figure 7B*). These findings revealed that brown adipocytes promote an abundance of Ly6C$^{low}$ monocytes by inducing their proliferation. Interestingly, this replicative behavior of BATA-CM-treated of Ly6C$^{low}$ monocytes, which we showed above are more oxidative than Ly6C$^{high}$ monocytes, is consistent with recent observations that proliferative cells have higher glucose consumption and oxidative phosphorylation relative to quiescent cells (*Yao et al., 2019*).

To buttress the above results, we also assessed the colony forming potential of in vitro-generated monocytes that received pre-treatment of WATA-CM or BATA-CM, to assess the potential for expansion of progenitor cells present in the samples. Colonies were identified as BFU-E (giving rise to erythroid cells); CFU-GEMM (giving rise to large mixed cultures of granulocyte, erythroid, macrophage, megakaryocyte; also, known as CMP); CFU-G (giving rise to granulocytes) or CFU-M (giving rise to macrophages). BATA-CM promoted growth of CMP/CFU-GEMM cultures over control Media (**p<0.01 Two-way ANOVA, Tukey's multiple comparisons, *Figure 7C*) and biased granulocyte/macrophage progenitors (widely known as GMP) towards macrophage over granulocyte differentiation,

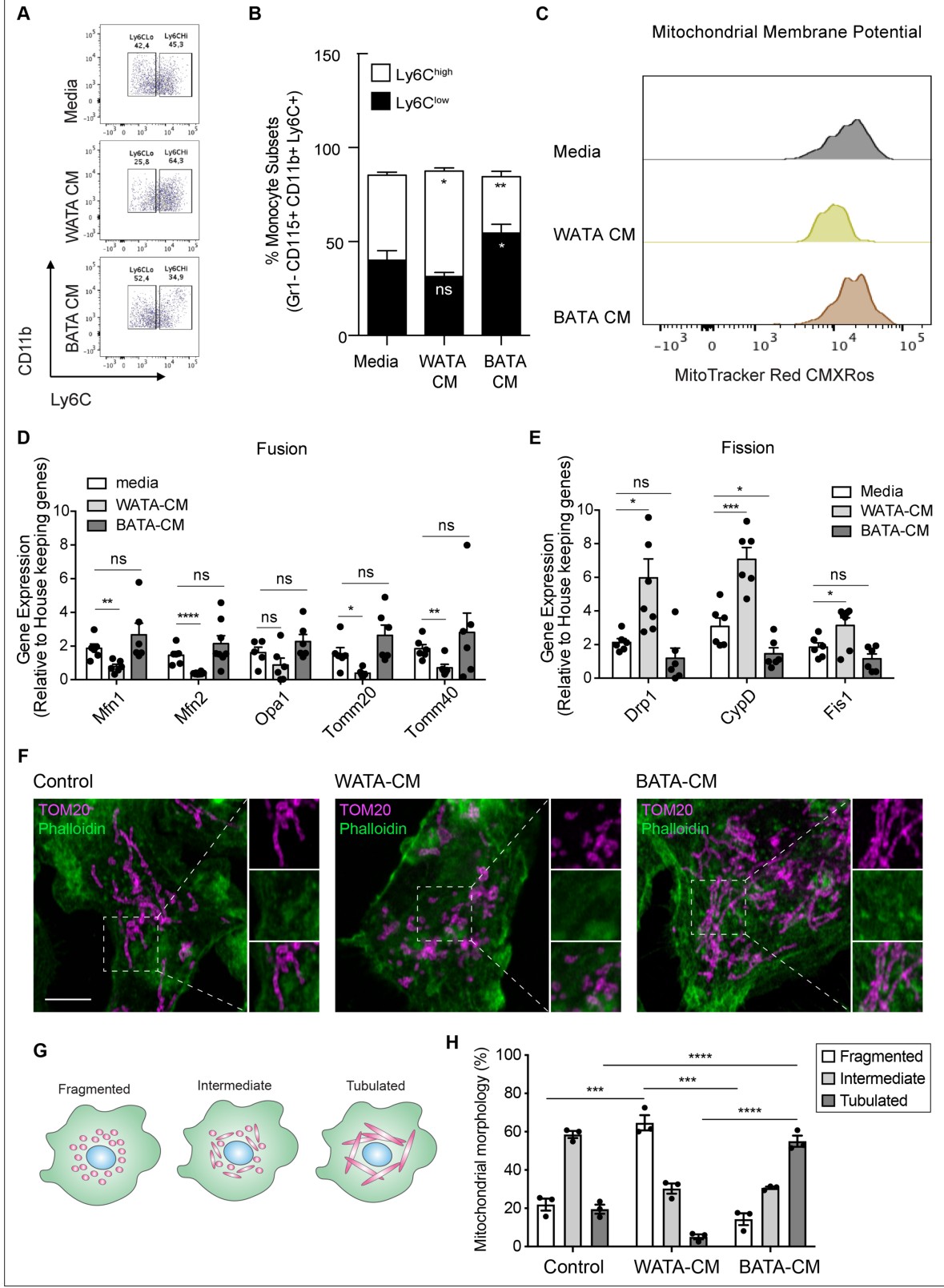

**Figure 6.** *Bona fide* white adipocytes favoured Ly6C^high monocytes preponderance, blunted monocyte mitochondrial membrane potential, and induced mitochondrial fragmentation in vitro. Naïve BM-derived monocytes were treated with CM from isolated *bone fide* white and brown adipocytes from EWAT and inter-scapular BAT for 18 hr. (**A**) Flow cytometry dot plots detecting the effects of 18 hr of treatment with media alone, pooled WATA-CM and BATA-CM from three mice (n=3 technical replicates each time with pooled CM from 3 mice) on monocyte subset abundance. (**B**) Quantification

*Figure 6 continued on next page*

*Figure 6 continued*

of the gates drawn for Ly6C$^{low}$ and Ly6C$^{high}$ monocytes (n=9 biological replicates of naïve monocytes treated as indicated). (**C**) Representative plot of MitoTracker Red CMXRos detected by flow cytometry in Ly6C$^{low}$ monocytes treated with media alone, pooled WATA-CM and BATA-CM for 18 h. (**D**) Gene expression of mitochondrial fusion genes *Mfn1*, *Mfn2*, *Opa1*, *Tomm20*, and *Tomm40* in naive BM-derived monocytes treated with control Media, pooled WATA-CM and BATA-CM (n=6 mice in each CM group and samples were analyzed in duplicates). (**E**) Gene expression of mitochondrial fission genes *Drp1*, *Ppid* and *Fis1* in naive BM-derived monocytes treated with control Media, WATA-CM and BATA-CM (n=6 mice in each CM group and samples were analyzed in duplicates). Results in (**B**), (**D**) and (**E**) are expressed as mean ± SEM, with one-way ANOVA used for statistical analysis: *p<0.05, **p<0.01 and ***p<0.001, ****p<0.0001 and ns = not significant p>0.05. Graphs in (**A**) and (**C**) were created using FlowJo software (Becton, Dickinson Company). (**F**) Super-resolution microscopy images of mitochondrial morphology via TOM20 staining in BM-derived monocytes treated with control Media, pooled WATA-CM and BATA-CM (representative of n=3 mice in each CM group). Scale bar = 5µm. Images were analyzed in Fiji (version 2.1.0/1.53 c). (**G**) Schematic illustration of mitochondrial morphologies as fragmented, intermediate, or tubulated. (**H**) Quantification of the percentage of monocytes with the indicated mitochondrial morphology, in each treatment condition. Values are from three independent experiments; n=3 mice in each CM group; about 40 cells counted for each condition, per experiment. Results are expressed as mean ± SEM. One-way ANOVA was used for statistical analysis. *p<0.05, **p<0.01 and ***p<0.001, ****p<0.0001 and ns = not significant p>0.05.

The online version of this article includes the following source data for figure 6:

**Source data 1.** *Bona fide* white adipocytes favoured Ly6C$^{high}$ monocytes preponderance, blunted monocyte mitochondrial membrane potential, and induced mitochondrial fragmentation in vitro.

relative to control Media (**p<0.01) or WATA-CM (*p<0.05 or ***p<0.001, *Figure 7C*). The total numbers of colonies that grew after 7–10 days of culturing of each pre-treated cohort of monocytes were not different across the three treatments, although trended upwards with BATA-CM (*Figure 7D*). These results indicated that while BATA-CM promoted expansion of selected populations (CMP/CFU-GEMM and CFU-M), consistent with the increase in BrdU incorporation shown above, WATA-CM was without effect relative to control Media.

The above findings suggest that while the proportional increase in Ly6C$^{low}$ monocytes induced by BATA-CM involves cell proliferation, the proportional increase in Ly6C$^{high}$ monocytes induced by WATA-CM does not. As a complementary approach, BM cells were analyzed by flow analysis by first gating for the Ly6C$^{low}$ or Ly6C$^{high}$ monocyte subsets (*Figure 2—figure supplement 1A*) and then analyzing for the presence of monocyte progenitors within the monocyte subsets (*Figure 2—figure supplement 1D*). Illustrated are the GMP, CMP and MEP gates in the monocyte subsets (*Figure 7E*). GMP progenitor cells were essentially the only progenitors detected by this approach in the Ly6C$^{low}$ monocyte subset, and they represented 140 GMP per 1000 Ly6C$^{low}$ cells (*Figure 7E*). During the incubation time of 18 hr with conditioned medium, we anticipate the progenitors could theoretically undergo only one doubling and therefore unlikely to account for the full changes in Ly6C$^{low}$ cell numbers produced by WATA-CM.

Collectively, the results in *Figure 7C–E* indicate that WATA-CM treatment did not result in an appreciable expansion of progenitor cells or colony formation. Therefore, alternative mechanisms were explored that might contribute to the WATA-CM induced shift towards Ly6C$^{high}$ monocyte preponderance, particularly the possible conversion of one subset into the other. It is known that Ly6C$^{high}$ monocytes give rise to Ly6C$^{low}$ monocytes in mouse blood and BM, and this is paralleled by the human homologous CD14 and CD16 monocyte subsets (*Kratofil et al., 2017*; *Lessard et al., 2017*; *Patel et al., 2017*). However, while there has been speculation about the conversion of Ly6C$^{low}$ to Ly6C$^{high}$ monocytes, there is little experimental evidence for this phenomenon. Hence, we contemplated the possibility that BATA-CM may also cause Ly6C$^{high}$ to Ly6C$^{low}$ monocyte conversion, and especially whether WATA-CM might cause Ly6C$^{low}$ to Ly6C$^{high}$ monocyte conversion, however unprecedented. To this end, we collected each FACS-sorted, homogenous Ly6C$^{low}$ and Ly6C$^{high}$ monocyte populations and, once isolated, treated them with WATA-CM, BATA-CM, or control media for 18 hr. Following treatments, we evaluated the presence of the alternative monocyte subtype relative to that in the respective starting homogenous populations. Surprisingly, no conversion from Ly6C$^{high}$ (red) to Ly6C$^{low}$ (gray) monocytes was observed in any condition (*Figure 8A–F*), but, remarkably, there was a switch of the FACS sorted Ly6C$^{low}$ into Ly6C$^{high}$ monocytes following exposure to WATA-CM (*Figure 8B and E*). While the FACS-sorted starting cells were all Ly6C$^{low}$ monocytes, following the 18 hr WATA-CM treatment the monocyte population consisted of 49.7% Ly6C$^{high}$ monocytes (and correspondingly only Ly6C$^{low}$ monocytes 50.3%) (*Figure 8E*). This was in stark contrast to all other isolated cell and CM treatment combinations tested, which gave rise to a maximum of 16.3% of the opposite monocyte subset compared to the starting population. Taken together, the results from *Figures 7 and 8* endorse

**Table 1.** List of Antibodies used for flow cytometry analysis.

| Markers | Fluorophore | Manufacturer | Clone | Catalog Number |
|---------|-------------|--------------|-------|----------------|
| CD11b | APC-eFluor 780 | eBioscience | M1/70 | 47-0112-82 |
| Ly6C | PE-Cy7 | Biolegend | HK1.4 | 128018 |
| Gr-1 | Alexa Fluor 700 | eBioscience | RB6-8C5 | 56-5931-82 |
| CX3CR1 | Brilliant Violet 421 | Biolegend | SA011F11 | 149023 |
| CCR2 | Alexa594 | R&D Systems | 475301 | FAB5538T |
| CD115 | APC | eBioscience | AFS98 | 17-1152-82 |
| F4/80 | PerCP-Cy5.5 | BioLegend | BM8 | 123128 |
| CD11c | FITC | eBioscience | N418 | 11-0114-81 |
| CD34 | FITC | BD Biosciences | RA1434 | 553733 |
| CD16/32 | APC-R700 | BD Biosciences | 2.4G2 | 565502 |
| CD45.2 | BV650 | BD Biosciences | 30-F11 | 563410 |
| CD150 | PEPCP710 | invitrogen | mShad150 | 46-1502-82 |
| CD117 | APC/Fire 750 | BioLegend | 2B8 | 105838 |
| SCA1 | PE-Cy7 | Invitrogen | D7 | 25-5981-82 |
| CD48 | BV605 | BioLegend | MM48-1 | 103441 |
| CD3e* | PE | BD Biosciences | 17A2 | 555725 |
| B220* | PE | BD Biosciences | RA3-6B2 | 553089 |
| Ly6G* | PE | BD Biosciences | 1A8 | 551461 |
| CD11b* | PE | BD Biosciences | M1/70 | 553311 |
| CD11c* | PE | BioLegend | N418 | 117308 |
| Ter119* | PE | BD Biosciences | TER-119 | 553623 |
| CD127* | PE | eBioscience | A7R34 | 12-1271-82 |
| NK1.1* | PE | BD Biosciences | PK138 | 553165 |

*LINEAGE is a cocktail of antibodies containing CD3e, B220, Ly6G, CD11b, CD11c, Ter119, CD127, NK1.1 with the same fluorophore to exclude all terminally differentiated cells.

the novel possibility of Ly6C$^{low}$ into Ly6C$^{high}$ monocyte conversion induced by secreted factors from white adipocytes.

## Discussion

In this study, we demonstrate a temporal correlation between HFD-induced whitening of BMA and a BM myeloid lineage bias towards Ly6C$^{high}$-bearing monocytes in vivo. The latter presented a more glycolytic metabolism underpinned by a reduction in mitochondrial membrane potential, a rise in mitochondrial fission genes and a reduction in mitochondrial fusion genes. Causatively, we show that CM from BMA of HFD-fed mice promotes the generation of Ly6C$^{high}$ monocytes ex vivo. Phenocopying the effect of BMA derived from HFD-fed mice, bona fide WATA favoured a Ly6C$^{high}$ monocyte preponderance, while BATA promoted a dominance of Ly6C$^{low}$ monocytes. Interestingly, these differential influences by the respective adipocytes are induced by distinct cellular pathways. Mechanistically, white adipocytes induced a shift in monocyte subset proportion, from Ly6C$^{low}$ to Ly6C$^{high}$, suggesting a possible and novel conversion of one subset into the other, while brown adipocytes favoured Ly6C$^{low}$ predominance by inducing their proliferation. These changes in monocytes were underpinned by alterations in the monocytes metabolism, as WATA blunted the monocyte's mitochondrial membrane potential and OCR. Overall, our findings expose a regulation of monocyte subsets directly influenced

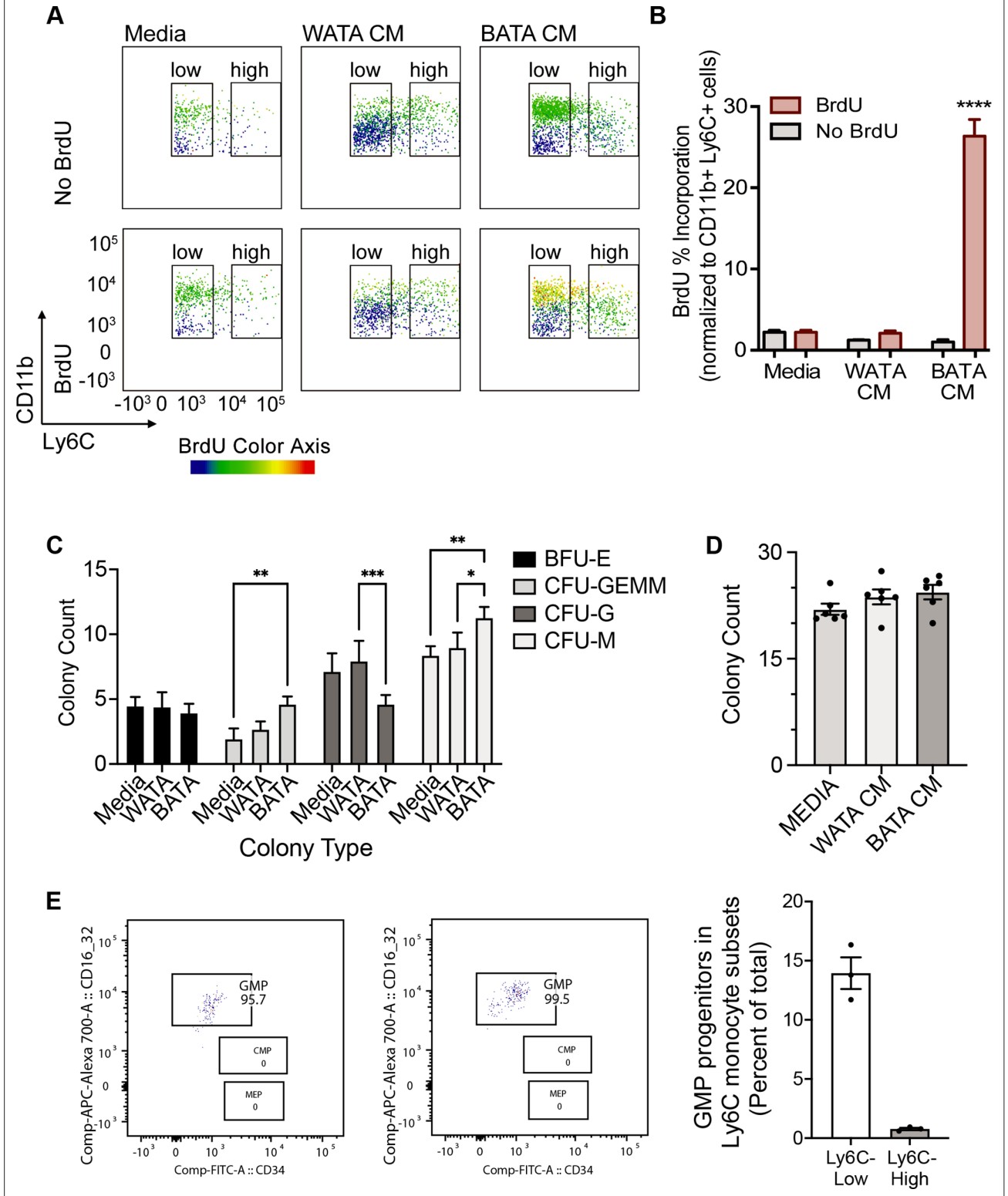

**Figure 7.** Brown adipocytes favour Ly6C[low] monocytes through progenitor cell differentiation and proliferation. (**A**) BM-derived, in vitro-generated monocytes were treated with media alone, WATA-CM or BATA-CM in the presence of the pyrimidine analogue BrdU to detect cellular proliferation. (**B**) Quantification of BrdU abundance in Ly6C[low] and Ly6C[high] monocyte subsets with the treatments in panel A, was determined by flow analysis (yellow to red signal, see Methods). Results are expressed as mean ± SEM, n=3–6 individual mice per group. One-way ANOVA test was used for statistical analysis, ****p<0.0005. (**C, D**) Colony Forming Unit assays were performed as described in Methods using six independent cultures of BM-derived, in

*Figure 7 continued on next page*

*Figure 7 continued*

vitro-generated monocytes (n=6 individual mice), were treated with WATA-CM, BATA-CM or control adipocyte media (Media) for 18 h prior to mixing with methylcellulose-based media and plating at 20,000 cells/ml to 6-well plates, in triplicate. (**C**) BATA-CM promoted CFU–GEMM cultures over Media alone (**p<0.01 Two-way ANOVA, Tukey's multiple comparisons) and skewed granulocyte/macrophage progenitors towards macrophage (CFU-M) over granulocyte (CFU-G) differentiation, relative to Media alone (**p<0.01) or WATA-CM (*p<0.05 or ***p<0.001). (**D**) The counts of the different colonies for each treatment (Media, WATA-CM or BATA-CM) were totaled and plotted as the mean ± SEM (n=6 individual mice). There was no statistical difference between the media treatments. (**E**) Isolated BM cells were processed for flow analysis to determine the progenitor cell populations present in the Ly6C^low or Ly6C^high monocyte subsets. The percent GMP ratio was calculated relative to the total cells detected in either monocyte subset as the mean ± SEM (isolated from n=3 individual mice). Flow analysis panels in A and E were created using FlowJo software (Becton, Dickinson Company).

The online version of this article includes the following source data and figure supplement(s) for figure 7:

**Source data 1.** Brown adipocytes favour Ly6C^low monocytes through progenitor cell differentiation and proliferation.

**Figure supplement 1.** HSC and their progenitors following the white and brown CM treatment for 18 hr.

**Figure supplement 1—source data 1.** HSC and their progenitors following the white and brown CM treatment for 18 hr.

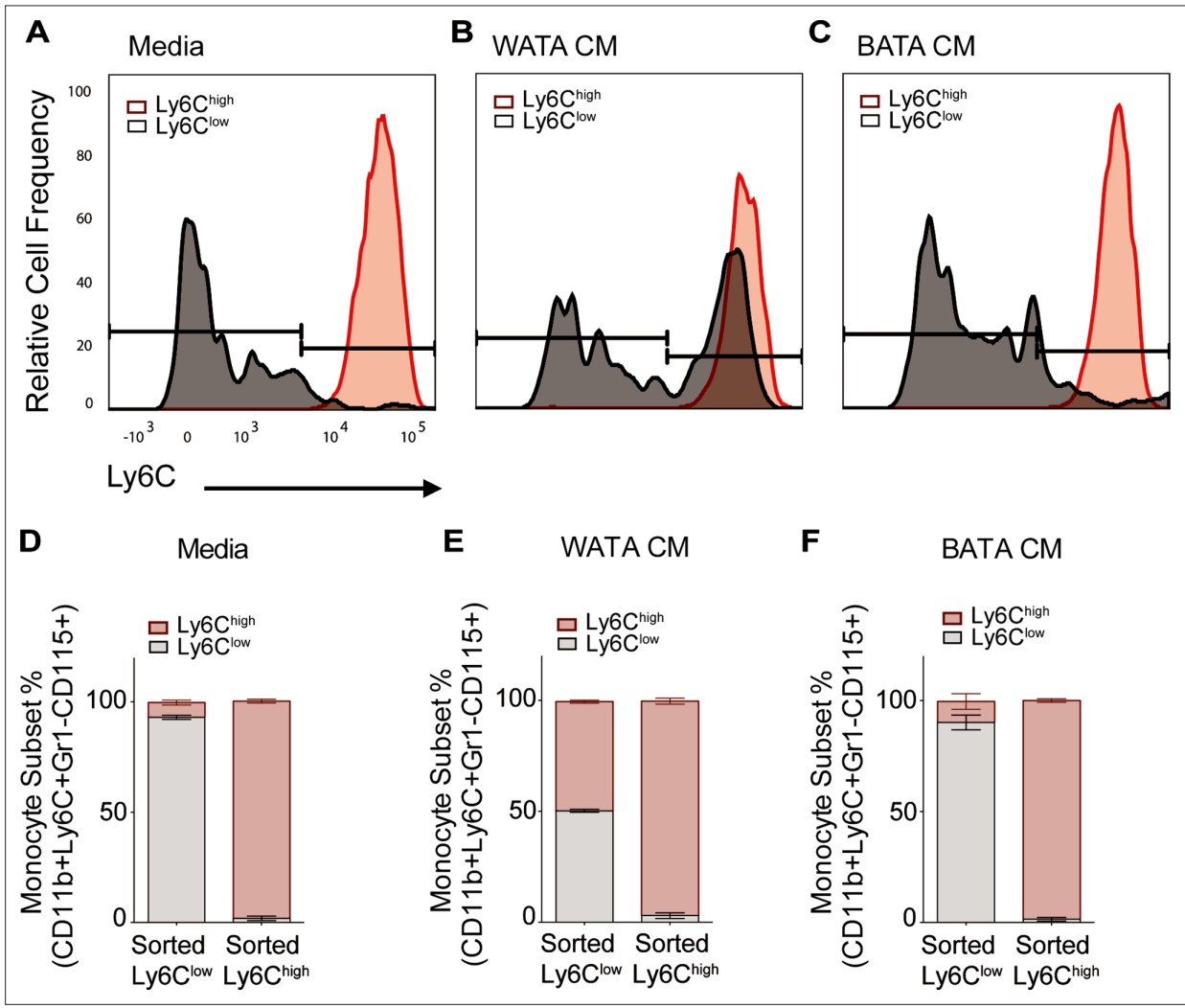

**Figure 8.** WATA-CM promotes Ly6C^low towards Ly6C^high shift in BM-derived monocytes. Representative plots of FACS-sorted Ly6C^low (gray) and Ly6C^high (red) monocytes from control mice were independently treated with (**A**) Media alone, (**B**) WATA-CM, or (**C**) BATA-CM (n=6 mice per FACS-sorted monocyte group) for 18 hr after which the presence of the alternate monocyte subset was assessed via flow cytometry. (**D–F**) Quantification of Ly6C^low and Ly6C^high gates in panels A-C. Flow cytometry panels A-C were created using FlowJo software (Becton, Dickinson Company).

The online version of this article includes the following source data for figure 8:

**Source data 1.** WATA-CM promotes Ly6C^low towards Ly6C^high shift in BM-derived monocytes.

by adipocytes, whether from the BM or from AT. Based on the high abundance of their defining marker, Ly6C, as well as of CCR2, plus their overall metabolic phenotype, we refer to this population as Ly6C$^{high}$ monocytes. Future in-depth analysis should define the nature of this interesting monocyte group. Along with the observation of dynamic whitening of BMA during HFD, our results contribute to the emerging evidence that BMA are not mere space fillers, rather they are important contributors to cellular metabolism and hematopoietic lineage decisions (*Boroumand and Klip, 2020*). That these changes occur early on in the course of diet-induced obesity suggests that the BM is an early responder to a change in feeding status and that adipocytes play a significant role in directing monocyte subset skewing. Moreover, they provide evidence of novel mechanisms regulating the genesis of such skewing.

## BM as an early sensor and an active contributor to HFD-induced AT macrophage accumulation

It is now widely recognized that many factors contribute to the accumulation of macrophages in AT that includes the contribution of monocytes. There is mounting evidence for both HFD-induced EWAT monocyte chemo-attraction and for enhanced BM myelopoiesis, that work together to increase the monocyte infiltration and pro-inflammatory AT macrophage accumulation (*Singer et al., 2014*; *Trottier et al., 2012* and *Figure 1D–H*). *Singer et al., 2014* demonstrated that transplantation of BM from 18week HFD-fed mice into control recipients increased the number of invasive circulating monocytes and pro-inflammatory AT macrophages. These pioneering findings heralded the notion that HFD influences hematopoietic myeloid bias in the BM upstream of monocyte tissue invasion, thereby contributing to downstream systemic and peripheral tissue inflammation. Furthermore, HFD-driven chemoattracting signals from the homeostatically disrupted EWAT initiate tissue influx and depletion of invasive Ly6C$^{high}$ monocytes from the circulation, which in turn may acts as a signal for their increased production in the BM (*Kamei et al., 2006*; *Nagareddy et al., 2014*; *Nov et al., 2013*; *Renovato-Martins et al., 2017*).

In the present study, by analyzing chronological changes in EWAT and BM myeloid composition, we observed that total BM monocytes increase by 3 weeks of HFD, whereas the proportion of Ly6C$^{high}$ monocyte predominance was evident only at 8 weeks coinciding with a gain in AT macrophages that were also pro-inflammatory skewed. We did not observe statistical significance in HFD-dependent changes in myeloid progenitors, likely due to their scarcity since HSC make up only 0.01% of the entire BM nucleated cells (*Challen et al., 2009*). The rise in total monocytes observed at 3 weeks of HFD and prior to elevation of AT macrophages identifies the BM as an earlier-responding *locus* during HFD. Interestingly, the increase was only apparent in Ly6C$^{low}$ patrolling monocytes and not the Ly6C$^{high}$ tissue invasive monocyte subset that is known for tissue infiltration in chronic obesity. This might be explained by an initial non-specific upregulation of total myelopoiesis and an increased Ly6C$^{high}$ monocyte BM extravasation. Therefore, it appears that only Ly6C$^{low}$ monocytes are elevated in the BM at 3 weeks of HFD. By 8 weeks of HFD, we observed a shift in the monocyte subset proportion, favouring Ly6C$^{high}$ monocytes. Thus, by 8 weeks, the BM has adopted the 'typical' HFD-induced hematopoietic phenotype, in so far as it already shows the increased Ly6C$^{high}$ monocytes previously reported at the habitually studied 16–18 weeks of HFD. Of note, we did not observe significant gains in progenitor cells (CMP, GMP) at either 3, 8 or 18 weeks of HFD compared to control diet (*Figure 2—figure supplement 2*), whereas Singer et al. found these cells elevated at 16 weeks of HFD (*Singer et al., 2014*; *Singer et al., 2015*). However, in the latter study, at 6 weeks of HFD, they did not observe changes in progenitor populations, yet an increase in circulating Ly6C$^{high}$ monocytes was already observed (Figure 5C in *Singer et al., 2015*). Collectively, those key studies and ours suggest the possibility that progenitor proliferation may be dissociated from the time during HFD when the specification leading to the increase in proportion of the Ly6Chigh subset takes place.

By comparison, we detected a gain in pro-inflammatory AT macrophages by 8 weeks of HFD that is concurrent with the predominance of invasive Ly6C$^{high}$ monocytes in the BM. In contrast, the embryonic-derived tissue resident AT macrophage markers, *Timd4* and *Marco* (*Beattie et al., 2016*; *Gibbings et al., 2015*), were not elevated until 18 weeks of HFD. This brings to light the importance of the contribution of the BM-derived myeloid cell pool to early EWAT pathology and AT macrophage accumulation.

## Changes in BMA influence monocyte subset skewing

BMA are a distinctive member of the BM cellular suite that is sensitive to HFD effects (*Pham et al., 2020*). In lean mice, BMA present characteristics that are similar to those of beige or brown adipocytes in peripheral AT (*Cuminetti and Arranz, 2019*). With HFD, BMA undergo hyperplasia and hypertrophy triggered by PPARγ activation (*Rosen and MacDougald, 2006*). Furthermore, with obesity, adipocyte precursor mesenchymal stem cells tilt their lineage commitment towards adipogenesis (*da Silva et al., 2016*; *Bowers and Singer, 2021*). While BMA are now known to be a distinct adipocyte population, they express several characteristics of white and brown adipocytes (*Boroumand and Klip, 2020*; *Krings et al., 2012*; *Sulston and Cawthorn, 2016*; *Tratwal et al., 2020*). Brown adipocyte markers of BMA are reduced in obese mice and a parallel adoption of more white adipocyte markers has been observed (*Adler et al., 2014*; *Doucette et al., 2015*; *Krings et al., 2012*). The amalgamation of upregulated white adipocyte proteins and elevated fat storage capacity is known as BM adipocyte whitening. Obesity-induced gains in whole body adiposity was causally linked to elevated circulating CD11b$^+$ monocytes (*Takahashi et al., 2003*). However, the influence of BMA from either lean or obese mice on the expansion of CD11b$^+$ monocytes had not been previously investigated.

The limited spatial capacity of the BM interior as well as the proximity of the cellular lineages in the various BM niches lead us to anticipate that the obesity-linked BMA whitening would influence the monocytic lineage. Here, we found a temporal, positive link between BMA hypertrophy and monocytosis at 3 weeks of HFD. By 8 weeks of HFD, BMA whitening also positively correlated with the rise in tissue invasive Ly6C$^{high}$ monocytes in the BM.

As a proof-of-concept for a direct and distinctive influence of adipocytes on monocyte number and subsets, we show that exposure to soluble factors from *bona fide* white adipocytes favours Ly6C$^{high}$ monocyte preponderance while exposure to soluble factors from brown adipocytes promotes the preponderance of Ly6C$^{low}$ monocytes. Mechanistically, we discovered that white adipocytes induced a change in the proportion of Ly6C$^{low}$ towards Ly6C$^{high}$ monocytes, an unprecedented process that is not well understood at present. Although the Ly6C$^{low}$ population contained some level of multipotential progenitor cells, it is unlikely that these contribute in a major way to the switch in preponderance of Ly6C$^{high}$ monocytes in response to WATA-CM, since this population did not undergo changes in the proximal lineage-committed precursors (CFU-M) when assayed for colony forming unit potential, and further did not involve BrdU accrual (indicative of DNA replication and cell proliferation).

Brown adipocytes, on the other hand, increased the proliferation of Ly6C$^{low}$ monocytes, as evinced by the higher incorporation of BrdU into the monocyte DNA and the expansion of monocyte lineage-committed precursor CFU-M colonies when tested for colony formation. Interestingly, secretome studies have reported that BAT-derived VEGFA and VEGFB can stimulate proliferation in adipocyte progenitors and endothelial cells (*Bagchi et al., 2013*; *Leung et al., 1989*). Adipocyte-specific VEGFA transgenic mice were protected from the effects of HFD and presented altered macrophage polarization (*Elias et al., 2012*). Therefore, it is conceivable that a reduction in brown adipocyte characteristics of BMA also leads to a drop in the factors that would normally favour proliferation of Ly6C$^{low}$ monocytes. While the identification of the soluble unknown factor(s) responsible for the adipocyte-specific myeloid influence is outside the focus of the present study, we have laid the foundation for a future, comprehensive investigation of the molecular mediators involved.

## How do adipocytes skew myeloid lineage subsets?

While the designation of macrophages as 'M1 and M2' (of pro- or anti-inflammatory characteristics, respectively) represents the extremes of a wide spectrum of phenotypes, the 'M1' pro-inflammatory orientation has been linked to glycolytic activation while the 'M2' anti-inflammatory orientation has been linked to oxidative phosphorylation (*Castoldi et al., 2016*; *Kelly and O'Neill, 2015*). A similar glycolytic metabolic switch with obesity has been reported for CD8 pro-inflammatory T cells (*Buck et al., 2017*). Surprisingly, while pro-inflammatory pathways in monocytes are activated by hyperglycemic conditions (*Shanmugam et al., 2003*), the metabolic preference of the different monocyte subsets had not been explored. This is a compelling question, given that Ly6C$^{high}$ monocytes penetrate tissues and differentiate to pro-inflammatory 'M1' macrophages while Ly6C$^{low}$ monocytes enter tissues to promote healing and repair (*Geissmann et al., 2003*; *Lin et al., 2009*). Therefore, we hypothesized here that, in the context of metabolic inflammation, there is a change in the metabolic phenotype of monocyte subsets.

Our findings show that Ly6C$^{high}$ and Ly6C$^{low}$ monocytes had different metabolic preferences at baseline, with Ly6C$^{low}$ having higher mitochondrial membrane potential indicative of more active oxidative phosphorylation. Interestingly, HFD consumption lowered the mitochondrial membrane potential of both Ly6C$^{low}$ and Ly6C$^{high}$ monocytes, implicating a reduction in their mitochondrial oxidative capacity. This change aligned with a HFD-induced reduction in oxygen consumption by Ly6C$^{low}$ monocytes. Importantly, this difference was observed as early as 3 weeks and exacerbated by 8 and 18 weeks of HFD. The elevated lactate concentration measured in the BM at all HFD durations may be reflective of metabolic changes in cells of the BM cavity. Since we did not observe changes in the lactate levels in WATA-CM or BATA-CM, the higher lactate in BM of HFD-fed mice may not originate from changes in BMA.

Along with the change in monocyte subset proportions, we observed an increase in genes responsible for mitochondrial fragmentation and a decrease in those involved in mitochondrial elongation in monocytes from HFD-fed mice and treated with WATA-CM and BMA-CM. The physiological consequence of these changes is manifest by the striking mitochondrial fragmentation in monocytes treated with WATA-CM in comparison to control or BATA-CM treated monocytes. HFD is known to induce mitochondrial fission in a number of cells that correlates with insulin resistance. A recent report linked the two phenomena through *Drp1* action, as *Drp1* inhibition reversed HFD-induced insulin resistance (*Filippi et al., 2017*). Furthermore, obese and diabetic humans have reduced *Mfn2* expression in skeletal muscle. *Mfn2* positively correlates with insulin sensitivity (*Bach et al., 2005*), and *Mfn2* gene and protein are significantly downregulated in liver cells of HFD-fed rats (*Gan et al., 2013*). In our analysis, the expression of *Mfn2* negatively correlated with the abundance of Ly6C$^{high}$ monocytes, while, the expression of *Drp1* positively correlated with the BM abundance of Ly6C$^{high}$ monocytes. Overall, our study uncovers novel and distinct effects of adipocyte types on monocyte subset development that may occur through changes in mitochondrial fitness.

Given that the proportion of Ly6C$^{low}$ monocytes dropped in favour of a predominance of Ly6C$^{high}$ monocytes by 8 but not 3 weeks of HFD, it is possible that this subset change obeys a prior switch in their metabolic orientation. Supporting these observations, exposure of monocytes to white adipocyte CM also reduced their mitochondrial membrane potential. Furthermore, BMA-CM from 8-week HFD-fed mice also lowered monocyte mitochondrial membrane potential compared to control BMA-CM and favoured Ly6C$^{high}$ predominance. Mechanistically, these data suggest that white adipocytes induce a monocyte metabolic switch that may guide Ly6C$^{high}$ monocyte prevalence.

In conclusion, we demonstrate an interconnectivity of cellular networks within the BM, a vulnerability of the BM microenvironment to HFD, and a consequential myeloid lineage determination likely due to BMA whitening. Because of the vastly different metabolic and soluble factors between bona fide white and brown adipocytes, it is easy to imagine that phenotypic changes in BMA can vastly alter the BM microenvironment and the factors that surround hematopoietic cells. BMA whitening during HFD feeding contributes to myelopoiesis and predominance of invasive Ly6C$^{high}$ monocytes mediated by changes in monocyte metabolism.

# Materials and methods
## Animal procedures

Mouse protocols were approved by the animal care committee (Protocol #20011850 to S.E.G. and D.J.P., University of Toronto and #1000047074 to A.K., The Hospital for Sick Children). Three groups of 9-week-old male C57BL/6 mice were fed either low fat diet (10% kcal from fat; D12450Ji, Research Diets) or HFD (60% kcal from fat; D12492i, Research Diets) for 3, 8, or 18 weeks. Each duration of feeding involved at least hree mice within each dietary group, and the precise number of mice evaluated in each experiment is indicated in the figure legends. The feeding protocol was repeated with three independent cohorts of mice, initiated at different times. The graphs illustrated include data collected from the three cohorts. Logistically, not every mouse could be used for every type of experiment, and the number of mice involved is indicated in each figure. Mice were singly caged to avoid stress and weight inequalities induced by aggression of the dominant male to the subordinate male (*Kappel et al., 2017*). The cages were maintained at 22° C on a 12 hr light/dark cycle, the diets were replaced once per week and various cage enrichments were provided including switching the positioning of the cages throughout the study.

Body weight was measured twice per week. Upon completion of the respective feeding durations, mice were fasted for 4 hr, blood was drawn via tail vein and fasting blood glucose was measured with a glucometer. Mice were euthanized by cervical dislocation. Tissues were dissected and flash frozen in liquid nitrogen for RNA isolation. EWAT, blood, and BM were immediately processed for flow cytometry analysis.

## EWAT-derived stromal vascular fraction (SVF) isolation

EWAT-derived SVF was collected as previously described (*Orr et al., 2013*). Excised EWAT was rinsed in PBS, minced and digested with 0.47wv/mL Liberase solution (Millipore Sigma, Cat No 5401054001) shaking at 200 rpm for 20 min at room temperature. SVF was isolated using a 70 µm cell strainer, the enzymatic activity of Liberase was neutralized with DMEM containing HI-FBS for 5 min and centrifuged at 500×g for 10 min. The supernatant containing adipocytes was removed, the pellet was re-suspended in red blood cell lysis buffer for 5 min to remove erythrocytes and washed with PBS three times.

## BM histology

Extracted tibia and femur bones were fixed in 10% formalin for 48 hr, decalcified in 14% EDTA solution for 14 days, paraffin wax-embedded, and longitudinal sections were stained with H&E and Oil Red O (Millipore Sigma; Cat No O0625) to visualize BM adiposity. Fifty imaging fields per mouse femur were analyzed for the presence of adipocytes, assessed by blinded volunteers. Adipocyte area was calculated using ImageJ software version 1.53.

## BM cell fractionation

Following cervical dislocation, femurs and tibiae were dissected from the mice fed control or the HFD for 3, 8, or 18 weeks. Epiphyses were cut from one end of each bone, then bones were placed in 0.2 mL microtubes containing 18-gauge holes. Microtubes were placed inside 0.5 mL Eppendorf tubes containing 100 µL media or PBS. This complex was centrifuged at 15,000 x g for 10 s to extract BM in the pellet. BM supernatant was removed from the cellular pelleted fraction to be assessed by ELISA and used for lactate quantification. BM cells were used for flow cytometry analysis, fluorescence-activated cell sorting (FACS) to identify monocyte subsets and their gene expression, as well as for differentiation to BM-derived monocytes.

## Primary adipocyte isolation and culture to generate conditioned media

Epididymal white and interscapular brown AT were dissected in sterile conditions, the tissues underwent mechanical mincing and enzymatic digestion by collagenase type I (Worthington Biochemical Corp, Cat No LS004197) at 4 mg/mL shaking for 1 hr. The digested adipocytes were then filtered through a 200 µm cell strainer and centrifuged, washed twice and plated in adipocyte media: high glucose DMEM, 1% BSA, 0.5% FBS, and 1% Penicillin and Streptomycin for 24 hr. BM adipocytes were isolated from the floating fraction following BM isolation via centrifugation. The cells were counted and cultured in adipocyte media overnight. Tissues from three mice were collected and the isolated cells were pooled in each CM generating experiment.

## In vitro-generated BM-derived monocytes

Murine primary BM cells were plated at 500,000 cells/mL on Thermo Scientific Nunc Non-Treated 6-well plates (Cat No 1256680) with complete RPMI (Cat No 350–000-EL, Wisent BioProducts, Saint-Jean-Baptiste, QC, Canada) and supplemented with recombinant mouse M-CSF (Peprotech Inc Cat No 315–02) at 20 ng/mL for 5 days (*Francke et al., 2011*). Following 5 days of BM cell incubation on the ultra-low attachment surface plates, the suspended cells have the signature of mature monocytes and can be used for experimentation. These BM-derived differentiated monocytes were then collected, washed with RMPI, centrifuged and re-suspended with various adipocyte CM for 18 hr. Typically, when BM cells were collected three mice were independently processed at the same time. This process yielded three cohorts of differentiated monocytes that would each be treated with Media-, WATA- and BATA-CM.

## Flow cytometry

Following red blood cell lysis, SVF, blood and the BM (or In vitro-generated, BM-Derived monocytes) were washed with FACS buffer (PBS, 1% (v/v) heat-inactivated FBS, 0.2% (w/v) BSA), stained with Aqua Live/Dead Antibody (ThermoFisher, L34957) for 15 min in the dark, washed with FACS buffer again, blocked with anti-mouse CD16/32 antibody (eBioscience, 14-0161-86) for 15 min in the dark following a 30 min incubation with corresponding antibody for each tissue (see *Table 1* for list of antibodies). The samples were then fixed, read on LSR Fortessa cytometer (BD Biosciences) and data gated based on appropriate fluorescence minus one (FMO) controls, and analyzed on the FlowJo software version 10 (Becton, Dickinson & Company).

## Fluorescent activated cell sorting of monocytes

BM from two femurs and tibiae of 8- to 10-week-old chow-fed mice were processed as above and used in the isolation of monocyte subsets via fluorescent activated cell sorting using the FACS Aria IIu instrument. Gr1$^-$ CD115$^+$ CD11b$^+$ and Ly6C$^+$ surface markers were used in identifying total monocytes and two subsets were subsequently distinguished by the relative surface abundance (low and high) of Ly6C protein (*Figure 2—figure supplement 1*). Isolated Ly6C$^{low}$ and Ly6C$^{high}$ monocytes were then validated for positive staining by antibodies CX3CR1 and CCR2 (*Figure 2—figure supplement 1C*), respectively. Ly6C and Cd11b FMO controls shown in *Figure 2—figure supplement 1B* illustrated that the gates drawn for Ly6C$^{low}$ and Ly6C$^{high}$ were confidently identifying cells positive for both these markers.

## BrdU incorporation

The abcam protocol was followed to detect BrdU through an anti-BrdU antibody. In vitro-generated, BM-derived monocytes were labelled with 10 µM BrdU (ThermoFisher, 000103) during the 18 hr treatment at 37 °C. The cells were then washed with FACS buffer, surface markers were stained as described above and incubated in Foxp3 Fixation/Permeabilization Buffer (eBioscience, 00-5523-00) overnight. They were then treated with 30 µg of DNAse I.9 (Sigma, 260913) for 1 hr, stained with anti-BrdU antibody (abcam, ab6326) for 1 hr and then washed with FACS buffer in preparation to run on LSR Fortessa Cytometer (BD Biosciences).

## Quantitative PCR (qPCR)

RNA was extracted from cells or tissues using TRIzol (Life Technologies, Cat No 15596018), then cDNA synthesized using the SuperScript VILO cDNA kit (Life Technologies, Cat No 11754250) according to manufacturer's instructions. qPCR reactions were performed with predesigned TaqMan probes (Life Technologies) on a StepOne Plus Real-Time PCR System (Applied Biosystems-ThermoFisher). Duplicates of each cDNA sample were used for gene expression analysis and values were normalized to the average of a set of the following housekeeping genes: *Abt1*, *Hprt*, and *Eef2* by the $\Delta\Delta C_T$ method.

## Lactate quantification

The BM content was fractioned into hematopoietic and supernatant portions by centrifugation at 1600 rpm. The supernatant fraction was re-suspended in 500 µL of 1 X PBS (supplemented with Protease Cocktail Inhibitor, Sigma). One drop of the BM supernatant fraction was placed on the Blood Lactate Meter Lactate Plus (Nova Biomedical) for the quantification of its lactate content. The values were normalized to the undiluted BM isolated from the femur.

## Cell metabolism analysis

Oxygen consumption (OCR) and extracellular acidification rates were measured using a Seahorse XFe96 analyzer (Seahorse Bioscience). Seahorse 96-well cell culture plates were pretreated with 1 µg/cm$^2$ Corning Cell-Tak Cell and Tissue Adhesive (Fisher Scientific, Cat No CB40240) and FACS sorted Ly6C$^{low}$ or Ly6C$^{high}$ monocytes were seeded at 100,000 cells/well, allowed to equilibrate for 1 hr in XF base medium supplemented with 11 mM glucose, 1 mM sodium pyruvate and 2 mM Glutamine. Following baseline measurements, FCCP was injected at 1 mM and thereafter an injection of Antimycin A/ Rotenone. Cells were counted following the Seahorse assay to test for changes in cellular detachment between the conditions.

## Immunofluorescence staining for super-resolution microscopy

Monocytes were spun down onto coverslips and fixed with 4% paraformaldehyde for 15 min. Thereafter, the cells were rinsed with PBS and permeabilized with 0.1% Triton X-100 for 10 min. The cells were then blocked with 2% BSA in PBS at 4°C overnight. The cells were then incubated with the primary antibody, Tom20 (FL-145, Santa Cruz, sc-11415), in a 1:100 dilution in blocking buffer at 4°C overnight. Thereafter, the cells were incubated with Alexa Fluor 488 anti-rabbit secondary antibody (1:100) for 2 hr at room temperature. Cells were incubated with Alexa Fluor 568 Phalloidin (1:100) for 3 hr at room temperature. Coverslips were then mounted by ProLong Diamond (Thermo Fisher Scientific) and allowed to dry for 2 days. The images were acquired via Zeiss LSM880 Airyscan super resolution microscopy and the images were processed via Fiji software. The percentage of monocytes displaying each mitochondrial morphology (fragmented, intermediate, or tubulated) was scored as described previously (*Rambold et al., 2011*).

## Colony-forming unit assay

In vitro generated, BM-derived monocytes ($5 \times 10^5$ cell/ml) were treated with WATA-CM, BATA-CM or media, as indicated, then pelleted and suspended in Iscove's modified Dulbecco's medium to $2 \times 10^5$ cell/ml, diluted in MethoCult GF-M3434 medium and plated at 20,000 cells/35 mm well in a 6-well plate of low-adherence culture-ware (STEMCELL Technologies, Inc), in triplicate. The colonies that grew had distinct cell lineages and morphologies as defined in detail by the STEM-CELL manual for MethoCult GF-M3434 methylcellulose media. After 7–10 days, colonies were counted, using an inverted light microscope, and identified as: Burst-forming unit-erythroid (BFU-E), Colony-forming unit-granulocyte, erythrocyte, macrophage, megakaryocyte (CFU-GEMM), CFU-granulocyte (CFU-G) or CFU-macrophage (CFU-M), based on the STEMCELL Technologies detailed instructions. Of note, CFU-GEMM progenitors are also known as CMP, and GMP progenitors give rise to the CFU-G and CFU-M lineages (immediate precursors of monoblasts and monocytes) that form colonies in this assay (https://www.stemcell.com/hematopoietic-stem-and-progenitor-cells-lp.html).

## ELISA determinations

The BM content was fractioned into hematopoietic and supernatant portions by centrifugation at 1600 rpm. The supernatant fraction was re-suspended in 500 µl of 1 X PBS (supplemented with Protease Cocktail Inhibitor, Sigma). Total protein content was estimated by Pierce BCA Protein Assay Kit (ThermoFisher Scientific). Subsequently, mouse *TNFα*, *Il1β*, *Il6*, *Cxcl1*, and *Ccl2* ELISAs were carried out according to the manufacturers' instructions. All Mouse DuoSet ELISA kits were purchased from R&D systems. The individual protein contents were finally normalised to total protein content.

## Statistical analysis

Data are expressed as means ± SEM. An unpaired Student's T test was used to detect differences in data sets with one variable such as that in *Figure 2—figure supplement 2*. A one-way ANOVA with Tukey post-test was used to detect differences in data sets of two or more variables. Statistical significance was set at $p < 0.05$. Data analysis and statistics tests were done using GraphPad Prism version 6 (GraphPad Software), Flow Jo Software version 10.4 and figures were prepared using Adobe Illustrator CS6 version 16.0.

Correlative analysis statistics were conducted using the linear regression analysis via GraphPad Prism version 6 (GraphPad Software) testing whether slopes and intercepts are significantly different. The $r^2$ represents the goodness of the fit while the P values greater than 0.05 represent whether the slope of the line of best fit is significantly non-zero.

## Acknowledgements

We thank Dr. Spencer Freeman for helpful discussion. This work was supported by Canadian Institutes of Health Research (CIHR) Foundation Grants FND-143203 to AK and FDN-14333 to DJP. Studentship support for PB was from a Banting & Best Diabetes Centre-Novo Nordisk Graduate Studentship (2017- 2019) and a SickKids Research Training Centre Award (2019- 2020).

## Additional information

### Funding

| Funder | Grant reference number | Author |
| --- | --- | --- |
| Canadian Institutes of Health Research | FDN-143203 | Amira Klip |
| Canadian Institutes of Health Research | FDN-14333 | Dana J Philpott |
| Banting & Best Diabetes Centre-Novo Nordisk Graduate Studentship | 2017- 2019 | Parastoo Boroumand |
| SickKids Research Training Centre Award | 2019- 2020 | Parastoo Boroumand |

The funders had no role in study design, data collection and interpretation, or the decision to submit the work for publication.

### Author contributions

Parastoo Boroumand, Data curation, Formal analysis, Validation, Investigation, Methodology, Writing - original draft; David C Prescott, Formal analysis, Validation, Investigation, Methodology; Tapas Mukherjee, Validation, Investigation, Methodology; Philip J Bilan, Data curation, Validation, Investigation, Visualization, Methodology, Writing - review and editing; Michael Wong, Jeff Shen, Ivan Tattoli, Angela Li, Nancy Shi, Lucie Y Zhu, Zhi Liu, Investigation, Methodology; Yuhuan Zhou, Tharini Sivasubramaniyam, Investigation, Visualization, Methodology; Clinton Robbins, Resources, Methodology; Dana J Philpott, Resources, Supervision, Funding acquisition, Project administration, Writing - review and editing; Stephen E Girardin, Resources, Supervision, Project administration; Amira Klip, Conceptualization, Resources, Supervision, Funding acquisition, Writing - original draft, Project administration, Writing - review and editing

### Author ORCIDs

Lucie Y Zhu  http://orcid.org/0000-0002-1048-5377
Amira Klip  http://orcid.org/0000-0001-7906-0302

### Ethics

Mouse protocols followed the strictest protocols dictated by the Canadian Institutes of Health Research guidelines and were approved by the animal care committee (Protocol #20011850 to S.E.G. and 483 D.J.P., University of Toronto; and #1000047074 to A.K., The Hospital for Sick Children).

### Decision letter and Author response

Decision letter https://doi.org/10.7554/eLife.65553.sa1
Author response https://doi.org/10.7554/eLife.65553.sa2

## Additional files

### Supplementary files

• Transparent reporting form

### Data availability

All data generated or analysed during this study are included in the manuscript and supporting files.

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
