## [Editor Report]

This is an important study providing compelling observations describing the metabolic response of monocyte subsets in conjunction with crosstalk with adipocytes in the murine bone marrow during leanness and obesity. This work convincingly reports bone marrow adaptations to lipid signals and how obesity affects monocyte biology, recruitment and differentiation locally in the bone marrow. This study at the crossroad of metabolism and immunology will be of interest for a wide community interested in obesity and macrophage biology.

---

## [Decision Letter]

**Decision letter after peer review:**

Thank you for submitting your article "Bone marrow adipocytes drive the development of tissue invasive Ly6Chigh monocytes during obesity" for consideration by *eLife*. Your article has been reviewed by 2 peer reviewers, and the evaluation has been overseen by a Reviewing Editor and Carlos Isales as the Senior Editor. The reviewers have opted to remain anonymous.

Essential revisions:

1) An essential revision requirement is to provide stronger evidence to validate the claim that Ly6Chi monocytes are increased due to Ly6Clo monocyte conversion to Ly6Chi monocytes. See related comments in the recommendation for the authors.

2) In addition, the authors need to provide the number of times each experiment was performed, provide absolute number (normalize the cell number to the weight of the tissue) as well as include gating strategies for all cell types already at the start of the paper.

*Reviewer #1 (Recommendations for the authors):*

– Could the number of times each experiment was performed be added to the figure legends?

– Please include gating strategies for all cell types already at the start of the paper. Including strategies to find the different subsets as referenced in the text. When Cx3cr1 and CCR2 are used to confirm Ly6Clo and Ly6Chi subsets (Figure S4C) it would be good to show expression of both markers on both subsets for comparison.

– Could the yield of monocytes compared with other progenitor cells be quantified following the 5 day in vitro culture protocol (Figure 5F-H).

– Can the authors demonstrate their sorted Ly6Clo monocytes do not contain any earlier progenitor populations (Figure 7A-C)?

– Can an alternative gating strategy be proposed for Ly6Chi monocytes which avoids the need to use Ly6C eg with CCR2 that can allow a fairer assessment of Ly6C expression following 18 hours of culture (Figure 7A-C).

– Can isotype controls be included for Ly6C expression in Figure 7A-C.

– Can qPCR be performed on the different monocyte subsets before and after culture with WATA and BATA for prototypical Ly6Clo and Ly6Chi monocyte genes to further validate the conversion from Ly6Clo to Ly6Chi. (Figure 7A-C)

– Can figure 4 be repeated with sorted subsets of the BM monocytes to demonstrate the differences that are now only correlated with monocyte subset abundance?

– Can some quantification be provided for Figure 6F?

*Reviewer #2 (Recommendations for the authors):*

Several claims within the paper need further experimental data to be valid.

1. The authors only provide %s of the different assessed cell subsets. In light of the growth of tissue and the dynamics of adipose tissue growth, it would be important to normalize the cell number to the weight of the tissue.

2. Prior literature has identified several adipose tissue-resident macrophage subsets. the authors should use these markers and assess the heterogeneity of macrophages at their time points (Silva et al., 2019, JEM).

3. A central claim of the paper is that Ly6clow monocytes mature or convert to Ly6chigh monocytes. However, the authors fail to formally show this either with transfer experiments or in vitro spike-in assays. Furthermore, the authors do investigate the abundance of myeloid progenitors but not their potential. Therefore, in order to stake that claim, it is imperative to further investigate this phenotype with one of the aforementioned assays.

---

## [Author Response]

Essential revisions:1) An essential revision requirement is to provide stronger evidence to validate the claim that Ly6Chi monocytes are increased due to Ly6Clo monocyte conversion to Ly6Chi monocytes. See related comments in the recommendation for the authors.

This is indeed an important question. We had already documented the presence of precursors cells in the bone marrow cells from mice fed control and high fat diet, and also after ex vivo treatment with conditioned media from white and brown adipose tissue. In response to this suggestion, we have now added a new analysis of the precursor markers in FACS-identified Ly6Chi and Ly6Clo monocytes and have also evaluated the potential contribution of *replication* of progenitor cells. We detected a small amount of progenitor cells in the Ly6Clo preparation that could potentially contribute to the expansion of Ly6Chi cells in response to WATA. However, our new experiments of Colony Forming Assays and previous ones of BrdU incorporation failed to show an expansion in progenitor-derived cells and in DNA replication, respectively. We have nonetheless left open the possibility that progenitor cells contribute in part to the increase in Ly6Chi cells, but we sustain that even if it occurred it would not be sufficient to account for the entirety of the proportional change in Ly6Clo to Ly6Chi cells, and we provide evidence endorsing the novel conversion of Ly6Clo cells into Ly6Chi. We expand on these new observations in the detailed response below and the indicated pages in the revised manuscript. The specific changes are indicated within the Answers to the Reviewers, following this section.

2) In addition, the authors need to provide the number of times each experiment was performed, provide absolute number (normalize the cell number to the weight of the tissue) as well as include gating strategies for all cell types already at the start of the paper.

Throughout the manuscript, we highlight in yellow all statements regarding the number of times each experiment was performed and other similar information. In particular:

a) We have stated the number of experiments performed in each figure legend, and we also state in Methods (page 24) the following:

“Each duration of feeding involved at least 3 mice within each dietary group, and the precise number of mice evaluated in each experiment is indicated in the figure legends. The feeding protocol was repeated with three independent cohorts of mice, initiated at different times. The graphs illustrated include data collected from the three cohorts. Logistically, not every mouse could be used for every type of experiment, and the number of mice involved is indicated in each figure.”

As well on page 25 as follows:

“Fifty imaging fields per mouse femur were analyzed for the presence of adipocytes, assessed by blinded volunteers. Adipocyte area was calculated using ImageJ software version 1.53.”

And on page 26 as follows:

“Typically, when BM cells were collected for monocyte maturation, three mice were independently processed at the same time. This process yielded three cohorts of differentiated monocytes that would each be treated with Media-, WATA- and BATA-CM.”

If the comment to normalize the number of cells refers to cells of the bone marrow, their number cannot be related to the weight of the entire bone as this structure contains overwhelmingly more structural material and other cells. Experimentally, the cells are flushed out into the same amount of fluid each time, so there is no weight index to refer them to.

If this comment refers to monocytes/macrophages in adipose tissue, the majority of studies in this field report the % of a particular cell type in the Soluble Vascular Fraction in samples from each diet group, and some studies report the difference in cell number in the fat pad in each diet group. We have used the former (express the content of monocytes, macrophages or pro-inflammatory macrophages as % of live cells in the SVF). Note that we simply indicate these changes at different times during the diet as reference, as our goal was not to analyze any mechanistic changes within adipose tissue, but rather the focus of the study was on changes within the bone marrow, as described above. We hope that these explanations clarify our choice or reporting proportion of cells within each niche.

b) We now present the gating strategies at the beginning of the paper, both in the Methods section as well as in the new Figure 2 —figure supplement 1 (formerly Figure S4).

Reviewer #1 (Recommendations for the authors):– Please include gating strategies for all cell types already at the start of the paper. Including strategies to find the different subsets as referenced in the text. When Cx3cr1 and CCR2 are used to confirm Ly6Clo and Ly6Chi subsets (Figure S4C) it would be good to show expression of both markers on both subsets for comparison.

We appreciate this comment, and regret that unfortunately, we did not perform the cross-assessment of Cx3r1 in Ly6Chi and CCR2 in Ly6Clo cells.

– Could the yield of monocytes compared with other progenitor cells be quantified following the 5 day in vitro culture protocol (Figure 5F-H).

Our experiments, presented in Figure 7 —figure supplemental 1, show that the percent of CD117+ cells in the in vitro-generated monocytes account for only 4-5% of the total cell number after the 5-day differentiation protocol. Further, the 18h incubation with WATA-CM resulted only in a small increase in the myeloid progenitor CMP subset without any difference in the abundance of CD117+Sca1+ cells; CD117+ Sca1- cells or GMP progenitors (Figure 7 —figure supplemental S1A-C).

– Can the authors demonstrate their sorted Ly6Clo monocytes do not contain any earlier progenitor populations (Figure 7A-C)?– Can an alternative gating strategy be proposed for Ly6Chi monocytes which avoids the need to use Ly6C eg with CCR2 that can allow a fairer assessment of Ly6C expression following 18 hours of culture (Figure 7A-C).– Can isotype controls be included for Ly6C expression in Figure 7A-C.

The comment is well taken. However, when analysing Ly6C lo and hi subsets, we routinely pre-incubate the cell samples prior to flow analysis or sorting with Anti-CD16/32 (Fc γ III/II) to mitigate non-specific binding of all antibody reagents to the Fc γ receptors. In addition, we always run a Ly6C FMO control (antibody cocktail minus Ly6C) to clearly delineate a Ly6C negative signal from a Ly6Clo gate. Further, we use primary antibodies directly conjugated to a fluorophore. Together, we believe these steps reduce the additional requirement of an isotype control for each of our surface markers, including Ly6C.

– Can some quantification be provided for Figure 6F?

In attendance to this comment and have now repeated these experiments using samples from an additional 3 mice, treated their BM-derived in vitro-generated monocytes with WATA-CM and BATA-CM, and analyzed mitochondrial morphology in >50 cells per condition, using a binning tool to quantify fragmented, intermediate and elongated mitochondria (Rambold et al., 2011. Proc Natl Acad Sci U S A. 108(25):10190-5. doi: 10.1073/pnas.1107402108). The new results are presented in Figure 6F-H.

Reviewer #2 (Recommendations for the authors):Several claims within the paper need further experimental data to be valid.1. The authors only provide %s of the different assessed cell subsets. In light of the growth of tissue and the dynamics of adipose tissue growth, it would be important to normalize the cell number to the weight of the tissue.

We did not report the % of monocyte subsets in adipose tissue, rather in the bone marrow. The weight of the adipose tissue over time and across diets is given in figure 1. In figure 2, we next simply report the proportion of monocytes, or of macrophages, or of pro-inflammatory macrophages over time and with each diet. The point routinely communicated in the field is the increase in immune cells within the adipose tissue as it expands, and to our knowledge, it is usually not reported relative to the size of the expanding tissue. We simply indicate these changes at different times during the diet as reference, but do not analyze any mechanistic changes within adipose tissue, as the focus of our study was the changes in monocytes within the BM.

2. Prior literature has identified several adipose tissue-resident macrophage subsets. the authors should use these markers and assess the heterogeneity of macrophages at their time points (Silva et al., 2019, JEM).

We agree that a deep analysis of the cellular diversity and nuances of the adipose tissue and BM populations would be interesting to do, but we submit that this would constitute a stand-alone, large analysis that lies beyond the gist of our study, which was to explore the relationship between the changes in adipocytes within the bone marrow and the increase in Ly6Chi that happen with diet.

Regarding adipose tissue-resident macrophage subset, we had reported the levels of the resident macrophage markers Timd4 and Marco to distinguish resident from recruited macrophages (Figure 2). However, we did not measure also the markers reported by Silva et al., because our focus was not adipose tissue cells, rather the bone marrow monocyte subsets, Ly6Clo and Ly6Chi. The markers used by Silva et al., and other authors as well, could be used in the future to characterize adipose tissue immune cells using RNAseq across times and conditions, a goal clearly beyond the scope of our study. That type of future study could even include the bone marrow cells (or the muscle, or the liver) to provide a dynamic landscape of immune cells in the body under high fat feeding stress. The Reviewer’s comment is taken as an inspiration for that future analysis.

3. A central claim of the paper is that Ly6clow monocytes mature or convert to Ly6chigh monocytes. However, the authors fail to formally show this either with transfer experiments or in vitro spike-in assays. Furthermore, the authors do investigate the abundance of myeloid progenitors but not their potential. Therefore, in order to stake that claim, it is imperative to further investigate this phenotype with one of the aforementioned assays.

We have attended to this comment in two ways, as follows:

a) In new experiments, we have now analyzed BM cells by flow analysis for the presence of monocyte progenitors within FACS-identified Ly6C^low^ or Ly6C^high^ monocyte subsets. GMP progenitor cells were essentially the only progenitors detected by this approach in the Ly6C^low^ monocyte subset, and they represented 140 GMP per 1000 Ly6C^low^ cells (Figure 7E). During the incubation time of 18 h with conditioned medium, we anticipate the progenitors could theoretically undergo only one doubling and therefore unlikely to account for the overall changes in Ly6C^low^ cell numbers produced by WATA-CM.

These results complement our previous observations measuring the progenitors into (CMP, GMP, MEP) and HSC and MPP in bone marrow cells isolated from mice fed control and HFD (Figure 2 —figure supplement 2). This choice of progenitor and precursor markers emulates the ones used in the key study of Singer et al., (2014) who measured GMP and pre-GM (CMP). In our study, there were no gains in any of these progenitors in response to HFD. The difference is now discussed on pages 19-20, lines 431-438 of our revised manuscript, as follows:

“Of note, we did not observe significant gains in progenitor cells (CMP, GMP) at either 3, 8 or 18 weeks of HFD compared to control diet (Figure 2 —figure supplemental 2), whereas Singer et al., found these cells elevated at 16 weeks of HFD (Singer et al., 2014, Singer et al., 2015). However, in in the latter study, at 6 weeks of HFD, they did not observe changes in progenitor populations, yet an increase in circulating Ly6C^high^ monocytes was already observed. Collectively, those key studies and ours suggest the possibility that progenitor proliferation may be dissociated from the time during HFD when the specification leading to the increase in proportion of the Ly6C^high^ subset takes place.”

B) Further attending to the reviewer’s question, we have now performed Colony Forming Assays in BM in vitro-generated monocytes treated with WATA (new Figure 7C,D). WATA treatment failed to cause an expansion in progenitor-derived cells, whereas treatment with BATA, which we propose causes a proliferation of Ly6Clo cells, did. The new text and results on pages 15-16, lines 341-365 read as follows:

“To buttress the above results, we also assessed the colony forming potential of in vitro-generated monocytes that received pre-treatment of WATA-CM or BATA-CM, to assess the potential for expansion of progenitor cells present in the samples. Colonies were identified as BFU-E (giving rise to erythroid cells); CFU-GEMM (giving rise to large mixed cultures of granulocyte, erythroid, macrophage, megakaryocyte; also, known as CMP); CFU-G (giving rise to granulocytes) or CFU-M (giving rise to macrophages). BATA-CM promoted growth of CMP/CFU-GEMM cultures over control Media (**p<0.01 Two-way ANOVA, Tukey’s multiple comparisons, Figure 7C) and biased granulocyte/macrophage progenitors (widely known as GMP) towards macrophage over granulocyte differentiation, relative to control Media (**p<0.01) or WATA-CM (*p<0.05 or ***p<0.001, Figure 7C). The total numbers of colonies that grew after 7-10 days of culturing of each pre-treated cohort of monocytes were not different across the three treatments, although trended upwards with BATA-CM (Figure 7D). These results indicated that while BATA-CM promoted expansion of selected populations (CMP/CFU-GEMM and CFU-M), consistent with the increase in BrdU incorporation shown above, WATA-CM was without effect relative to control Media.

The above findings suggest that while the proportional increase in Ly6C^low^ monocytes induced by BATA-CM involves cell proliferation, the proportional increase in Ly6C^high^ monocytes induced by WATA-CM does not.

Collectively, those results indicate that WATA-CM treatment did not result in any appreciable expansion of progenitor cells or colony formation. Accordingly, alternative mechanisms had to be explored that might contribute to the WATA-CM induced shift towards Ly6C^high^ monocyte preponderance.”